# Prevalence of corneal findings and their interrelation with hematological findings in monoclonal gammopathy

**Mohammad Al Hariri[1], Markus Munder[2], Walter Lisch[1], Alexander K. Schuster[1], Eva-Marie Fehr[2], Björn Jacobi[3], Alexander Desuki[2], Andreas Kreft[4], Adrian Gericke[1], Norbert Pfeiffer[1], Joanna Wasielica-Poslednik[1] ***

**1** Department of Ophthalmology, University Medical Center of the Johannes Gutenberg-University Mainz, Mainz, Germany, **2** Department of Hematology, Oncology, and Pneumology, University Medical Center of the Johannes Gutenberg-University Mainz, Mainz, Germany, **3** Department of Gastroenterology, Hepatology, Diabetology, Endocrinology and Oncology, Klinikum Worms, Worms, Germany, **4** Institute for Pathology, University Medical Center of the Johannes Gutenberg-University Mainz, Mainz, Germany

* joanna.wasielica-poslednik@unimedizin-mainz.de

## Abstract

### Purpose

To determine prevalence of paraproteinemic keratopathy (PPK) among patients with monoclonal gammopathy (MG). To evaluate interrelation between corneal and hematological parameters in patients with PPK.

### Methods

Fifty-one patients with monoclonal gammopathy of undetermined significance (n = 19), smoldering multiple myeloma (n = 5) or multiple myeloma (n = 27) were prospectively included in this study. Best-corrected visual acuity, slit-lamp biomicroscopy, Scheimpflug tomography, in-vivo confocal laser scanning microscopy, optical coherence tomography and complete hematological workup were assessed.

### Results

We identified n = 19 patients with bilateral corneal opacities compatible with PPK. PPK was newly diagnosed in 13 (29%) of 45 patients with a primary hematological diagnosis and in n = 6 patients without previous hematological diagnosis. The most common form was a discreet stromal flake-like PPK (n = 14 of 19). The median level of M-protein (p = 0.59), IgA (p = 0.53), IgG (p = 0.79) and IgM (p = 0.59) did not differ significantly between the patients with and without PPK. The median level of the FLC κ in serum of patients with kappa-restricted plasma cell dyscrasia was 209 mg/l in patients with PPK compared to 38.1 mg/l in patients without PPK (p = 0.18). Median level of FLC lambda in serum of patients with lambda-restricted plasma cell dyscrasia was lower in patients with PPK compared to patients without PPK (p = 0.02).

**Data Availability Statement:** All relevant data are within the manuscript and its Supporting Information files.

**Funding:** The author(s) received no specific funding for this work.

**Competing interests:** The authors have declared that no competing interests exist.

## Conclusion

The PPK was mostly discreet, but its prevalence (29%) was higher than expected. Median level of the monoclonal paraprotein was not significantly higher in patients with PPK compared to patients without PPK. Our results suggest a lack of correlation between morphology and severity of the ocular findings and severity of the monoclonal gammopathy.

## Trial registration

German Clinical Trial Register: DRKS00023893.

## Introduction

The term "paraproteinemic keratopathy" (PPK) describes heterogeneous corneal opacity patterns caused by deposition of paraprotein in different corneal layers [1, 2]. The first case reports are dating back to the early 20th century. Blobner was the first to describe the crystalline deposits throughout the cornea associated with paraproteinemia in 1938 [3]. The first description of non-crystalline corneal deposits in connection with multiple myeloma (MM) was made in 1934 by Meesmann [4]. PPK may morphologically mimic some hereditary, degenerative or inflammatory corneal diseases [5]. Its prevalence has been described once, so far [6]. Bourne et al. found crystalline keratopathy in 1 of 100 patients with a confirmed diagnosis of multiple myeloma (MM) [6]. The morphology of the PPK can be very heterogeneous. Lisch et al. presented five morphological types of PPK in the first classification in 2012 [5]. The latest classification of PPK based on the morphology of corneal clouding was proposed in 2016 and distinguished between seventeen different forms of PPK [1].

Monoclonal gammopathy (MG) is defined as the presence of monoclonal immunoglobulins or their parts (free light- and/or heavy-chains) in blood and/or urine. MG is mostly due to a cytogenetically heterogeneous clonal plasma cell disorder but can be associated with any form of monoclonal B-cell disease. MM is almost always preceded by asymptomatic premalignant stages termed monoclonal gammopathy of undetermined significance (MGUS) und smoldering multiple myeloma (SMM) [7]. With a prevalence of 5.1% among persons over 50 years, MGUS is one of the most common premalignant disorders in Western countries [8]. MGUS has a risk of progression to MM of approximately 1% per year. MM accounts for around 10% of all malignant hematological diseases.

The ocular manifestations of MG can occur in virtually any structure of the eye and can be the first manifestation of the disease. They can appear as one of the extramedullary manifestations of the disease or as the first sign of an ineffective therapy [9, 10]. PPK can be associated with lymphoma, leukemia, cryoglobulinemia, rheumatoid arthritis or Waldenström macroglobulinemia [11].

The corneal stroma appears to be the most frequently affected corneal layer, followed by the epithelium and the subepithelium [11–13]. Endothelium and Descemet's membrane are rare locations of the paraprotein deposition [11, 14]. Crystalline paraprotein deposits may be visualized in the cornea using in vivo confocal laser scanning microscopy (IVCM) [15].

The term "monoclonal gammopathy of ocular significance" (MGOS) was proposed in 2019 for cases with visual impairment caused by PPK [16]. However, the hematological guidelines do not consider any form of PPK to be an end organ damage associated with MG, at present [17, 18].

It is unknown whether the presence and morphology of PPK correlate with the type of the paraprotein and with the severity of the hematological disorder.

The primary aim of the present study was to investigate prevalence of the PPK among patients with MG. The secondary aim was to investigate whether the type and severity of the MG determines the presence, morphology, and severity of the PPK. A further goal of our study was to raise the awareness of the occurrence of corneal involvement in hematological disorders. We hypothesize that the PPK is an underdiagnosed ocular condition and its prevalence among MG patients is higher than expected. Furthermore, we hypothesize that severity and morphology of PPK correlate with serum paraprotein level and type of MG.

## Methods

### Patients

This prospective observational cross-sectional clinical study was carried out in accordance with the Declaration of Helsinki. Ethics approval was obtained from the Ethics committee of Rhineland-Palatinate, Germany [vote no. 837.153.16 (10472)]. All patients were recruited and evaluated between September 2016 and October 2019 at the Department of Ophthalmology and at the Department of Hematology of the University Medical Center of the Johannes Gutenberg University Mainz. Written informed consent was obtained from all participants.

The study was registered in the German Clinical Trial Register: DRKS00023893 on 11[th] January 2021. The authors confirm that all ongoing and related trials for this drug/intervention are registered.

The inclusion criteria were: men and women over 18 years with the diagnosis MGUS, SMM or MM, capable of giving consent to participate in the study. The exclusion criteria included: condition after bilateral refractive corneal surgery; corneal and intraocular inflammation; patients with non-measurable M-protein in urine and serum, asecretory multiple myeloma and plasma cell leukemia, previous systemic therapy.

We aimed to recruit a maximum of consecutive patients meeting our inclusion and exclusion criteria in the period of maximal 3 years.

We recruited 45 patients at the Department of Hematology with a confirmed diagnosis of monoclonal gammopathy (MGUS, SMM or MM) and without any known corneal pathology. The inform consent and the ophthalmological examination followed the complete hematological workup and was performed within 3 months after the diagnosis of MG. The primary aim of our study—prevalence of PPK—was investigated in these patients.

We recruited additionally 6 patients at the Department of Ophthalmology with a primary suspicion of PPK. These patients underwent a hematological workup within 3 months after the ophthalmological diagnosis. We included them in the study (but not into the prevalence assessment) as soon as the hematological diagnosis of MG was confirmed.

MGUS is defined as: serum monoclonal protein < 30 g/l, clonal bone marrow plasma cells <10%, and absence of an end organ damage defined as hypercalcemia, renal insufficiency, anemia bone lesions (CRAB features) and special biomarkers of malignancy (SLiM features) or amyloidosis that can be attributed to the plasma cell proliferative disorder. SMM diagnosis must meet both conditions: Serum monoclonal protein (IgG or IgA) ≥30 g/L or urinary monoclonal protein ≥500 mg/24 h and/or clonal bone marrow plasma cells 10–60%; and: absence of amyloidosis or myeloma defining events (like CRAB features or special biomarkers of malignancy). MM is defined as serum monoclonal protein (IgG or IgA) ≥30 g/l or urinary monoclonal protein ≥500 mg/24 h and/or clonal bone marrow plasma cells ≥10%; and one or more of myeloma defining events (CRAB features or special biomarkers of malignancy, which indicate the high level of activity). The special biomarkers of malignancy (SLiM features) are

sixty percent (60%) or greater of bone marrow plasma cells, free light-chain ratio of 100 or greater in serum and MRI with more than one focal lesion. Also a renal failure due to light-chain cast nephropathy is considered a myeloma-defining event [7]. The term monoclonal gammopathy of clinical significance (MGCS) is used to define monoclonal gammopathies with a related important organ involvement and followed the term monoclonal gammopathy of renal significance (MGRS) [18].

## Hematological evaluation

The hematological evaluation was performed in the central laboratory of the University Medical Center of the Johannes Gutenberg University Mainz according to the clinical routine of the Department of Hematology. They included following laboratory tests: differential blood count, electrolytes, serum creatinine, uric acid, serum urea, alanine transaminase (ALT), aspartate transaminase (AST), alkaline phosphatase (ALP), gamma-glutamyl transferase (GGT), total bilirubin, C-reactive protein (CRP), lactate dehydrogenase (LDH), immunofixation in serum, immunoglobulins (Ig) G, A, M, D, and E, kappa (κ)—and lambda (λ) light-chains (LC) in serum, kappa/lambda ratio, free kappa light-chains (FLC κ), free lambda light-chains (FLC λ), free kappa/lambda light-chains ratio, M gradient, serum protein electrophoresis, serum albumin, beta-2-microglobulin, international normalized ratio (INR), activated partial thromboplastin time, total fibrinogen; 24-hour urine collection included: creatinine clearance, albumin, kappa and lambda light-chains and immunofixation; bone marrow puncture including histology, cytology, flow cytometry and cytogenetics; low-dose whole-body computed tomography. All laboratory tests were performed in the central laboratory of the University Medical Center of the Johannes Gutenberg-University Mainz.

## Ophthalmological evaluation

The ophthalmological evaluation performed in all study participants included respectively: autorefractometer (model AR-360A, Nidek Co, Wetzlar, Germany), best corrected visual acuity (BCVA) measured with Snellen charts; slit lamp biomicroscopy, fundoscopy, Scheimpflug tomography (Pentacam HR Typ 70900, Fa. Oculus, Wetzlar, Germany), optical coherence tomography (OCT, Spectralis: anterior segment module and macular volume scan, Heidelberg Engineering, Germany) of the cornea and macula, Goldmann applanation tonometry and IVCM (HRT2-04318 Rostock Cornea Model, Heidelberg Engineering GmbH, Germany).

We converted Snellen to logMAR for analysis. We performed the slit lamp examination in order to rule out corneal opacities or deposits. The fundoscopy was carried out to rule out retinal involvement in MG. The central corneal thickness and densitometry were assessed by Scheimpflug tomography, and the location of the corneal deposits was shown by means of anterior segment OCT. Retinal fluid was assessed by macular OCT. Ocular findings regarding ocular involvement in MG were the same in both eyes of all patients. Thus, mean values of right and left eyes of an individual subject was computed and analysis of visual acuity, corneal thickness, and densitometry was computed between involved vs. uninvolved subjects.

## Statistical analysis

Descriptive analysis included computation of absolute and relative frequency for categorical data. For continuous data, mean and standard deviation were calculated for normally distributed data, otherwise median and interquartile range. Prevalence estimates including 95% confidence interval was computed for PPK among patients with initial diagnosis of MG. Serum level of immunoglobulins and of light-chain levels were compared between patients with and without PPK for each type of MG using the Mann-Whitney-U test. Statistical analysis was

conducted by using R (Version 4.0.0) R Core Team (2021). R: A language and environment for statistical computing. R Foundation for Statistical Computing, Vienna, Austria. URL https://www.R-project.org/.

## Results

One hundred and two eyes of n = 51 patients, aged 63.3 ± 10.6 years (range 40–90), of which n = 29 (56.9%) were male, were included in this prospective study. The patients' hematological diagnoses included: MGUS (n = 19), SMM (n = 5), and MM (n = 27). Forty-five patients with a primary diagnosis of MG were recruited at the Department of Hematology. Six patients with a primary suspicion of PPK and a secondary diagnosis of MG were recruited at the Department of Ophthalmology.

We summarize the patients' demographic and clinical characteristics in Table 1.

### Prevalence of PPK

Forty-five consecutive patients with primary diagnosis of MG without any known corneal pathology were transferred from the Department of Hematology to rule out the PPK. We observed bilateral corneal opacities compatible with PPK on slit lamp examination in both eyes of 13 patients. This results in an overall prevalence of 28.9%.

One of 45 patients presented crystalline keratopathy on slit-lamp examination. Therefore, the prevalence of crystalline PPK in our study was 2%. However, the IVCM examination revealed hyperreflective needle-like crystalline stromal deposits in all corneas with diagnosis of PPK on the slit lamp examination.

We present the clinical characteristics of these patients in Table 2.

**Table 1. Demographic and clinical characteristics of all study participants regarding the hematological diagnosis (a) and the presence or absence of PPK (b).**

a

| MG—Form | n | male n = | mean age ± SD | female n = | mean age ± SD | PKK | n |
|---|---|---|---|---|---|---|---|
| MGUS | 19 | 14 | 66.9 ± 11.9 | 5 | 66.4 ± 13.2 | with | 8 |
| | | | | | | without | 11 |
| SMM | 5 | 2 | 72.5 ± 2.5 | 3 | 66.3 ± 8.2 | with | 4 |
| | | | | | | without | 1 |
| MM | 27 | 13 | 58.8 ± 10 | 14 | 63.2 ± 6.5 | with | 7 |
| | | | | | | without | 20 |
| All patients | 51 | 29 | 63 ± 11.9 | 22 | 63.9 ± 9.1 | with | 19 |
| | | | | | | without | 32 |

b

| Corneal status | n | gender | n | mean age ± SD | MGUS | | SMM | | MM | |
|---|---|---|---|---|---|---|---|---|---|---|
| | | | | | n | % | n | % | n | % |
| PPK | 19 | male | 9 | 66.6 ± 12.3 | 5 | 26.3 | 1 | 5.3 | 3 | 15.8 |
| | | female | 10 | 64 ± 6.4 | 3 | 15.8 | 3 | 15.8 | 4 | 21 |
| Non-PPK | 32 | male | 20 | 62.4 ± 10.9 | 9 | 28.1 | 1 | 3.1 | 10 | 31.2 |
| | | female | 12 | 63.8 ± 10.9 | 2 | 6,3 | - | | 10 | 31.2 |
| All patients | 51 | male | 29 | 63 ± 11.9 | 14 | 27.5 | 2 | 3.9 | 13 | 25.5 |
| | | female | 22 | 63.9 ± 9.1 | 5 | 9.8 | 3 | 5.8 | 14 | 27.5 |

SD—standard deviation, MG–monoclonal gammopathy, MGUS–monoclonal gammopathy of undetermined significance, SMM–smoldering multiple myeloma, MM–multiple myeloma, PPK–paraproteinemic keratopathy.

**Table 2. Clinical characteristics of 45 patients with primary diagnosis of monoclonal gammopathy.**

|  | PPK | | non-PPK | |
|---|---|---|---|---|
|  | n = 13 | 28.9% | n = 32 | 71.1% |
| MGUS | n = 3 | 6.7% | n = 11 | 24.4% |
| SMM | n = 3 | 6.7% | n = 1 | 2.2% |
| MM | n = 7 | 15.6% | n = 20 | 44.4% |

MGUS–monoclonal gammopathy of undetermined significance, SMM–smoldering multiple myeloma, MM–multiple myeloma, PPK–paraproteinemic keratopathy.

## Morphology of PPK

**Slit lamp.** We analyzed morphology of PPK among all patients included into the study (19 PPK cases from 51 included MG patients). The most common form of PPK was a diffuse stromal flake-like opacity (Fig 1A and 1B) observed in n = 14 patients (74%), followed by a peripheral superficial band-like PPK (Fig 1C) in 2 patients. One patient showed bilateral diffuse stromal opacity with cyst-like changes and lattice lines (stromal lattice-like PPK, Fig 1D). In one patient bilateral stromal punctiform crystalline-like deposits were found in the mid-peripheral corneal region (Fig 1E). One patient with biclonal IgG and IgA lambda type MM and hypercupremia showed PPK in the form of a central golden-brown discoloration of the pre-Descemet region (Fig 1F).

**IVCM.** IVCM revealed hyperreflective needle-like crystalline stromal deposits in all corneas with diagnosis of PPK on the slit lamp examination (Fig 2).

We found less pronounced hyperreflective point- and/or needle-like deposits in the deep stroma of 18 patients without any discernible opacities in the morphological examination of the cornea using the slit lamp.

## Clinical findings

Median BCVA (logMAR) in patients with PPK was 0.1 (range 0.8–0.0) and in patients without PPK 0.0 (range 1.3–0.0, p = 0.10). Median central corneal thickness in patients with PPK was 558 μm (range 499–759 μm) and in patients without PPK 556 μm (range 485–613 μm, p = 0.58). Median corneal densitometry measured with Scheimpflug tomography in patients with PPK was 36.4 (range 29–100) and in patients without PPK 35.6 (range 26.8–60.6; p = 0.15). Table 3 provides a tabular representation of the serological and ophthalmological findings.

Tonometry revealed normal intraocular pressure values in all included patients.

Fundoscopy and macula OCT revealed pathological findings in two patients. One patient with Waldenstrom's disease had bilateral macular involvement in form of subfoveal defects of the retinal pigment epithelium with cystoid macular edema and one patient with SMM type IgA λ had macular involvement in form of a focal detachment of the neurosensory retina with secondary choroidal neovascularization (CNV). We interpret these findings as paraproteinemic maculopathy.

## Hematological findings

Serologically, 11 of 19 patients with PPK (58%) had kappa-restricted plasma cell dyscrasia (7 with IgG, 2 with IgA and 2 with light-chain only), and 8 of 19 patients (42%) had lambda-restricted plasma cell dyscrasia (1 with IgG, 4 with IgA, 1 with both IgG and IgA and 2 with light-chain only). We summarize the serological MG-types and the morphological PPK-types in Table 4 and in Fig 3.

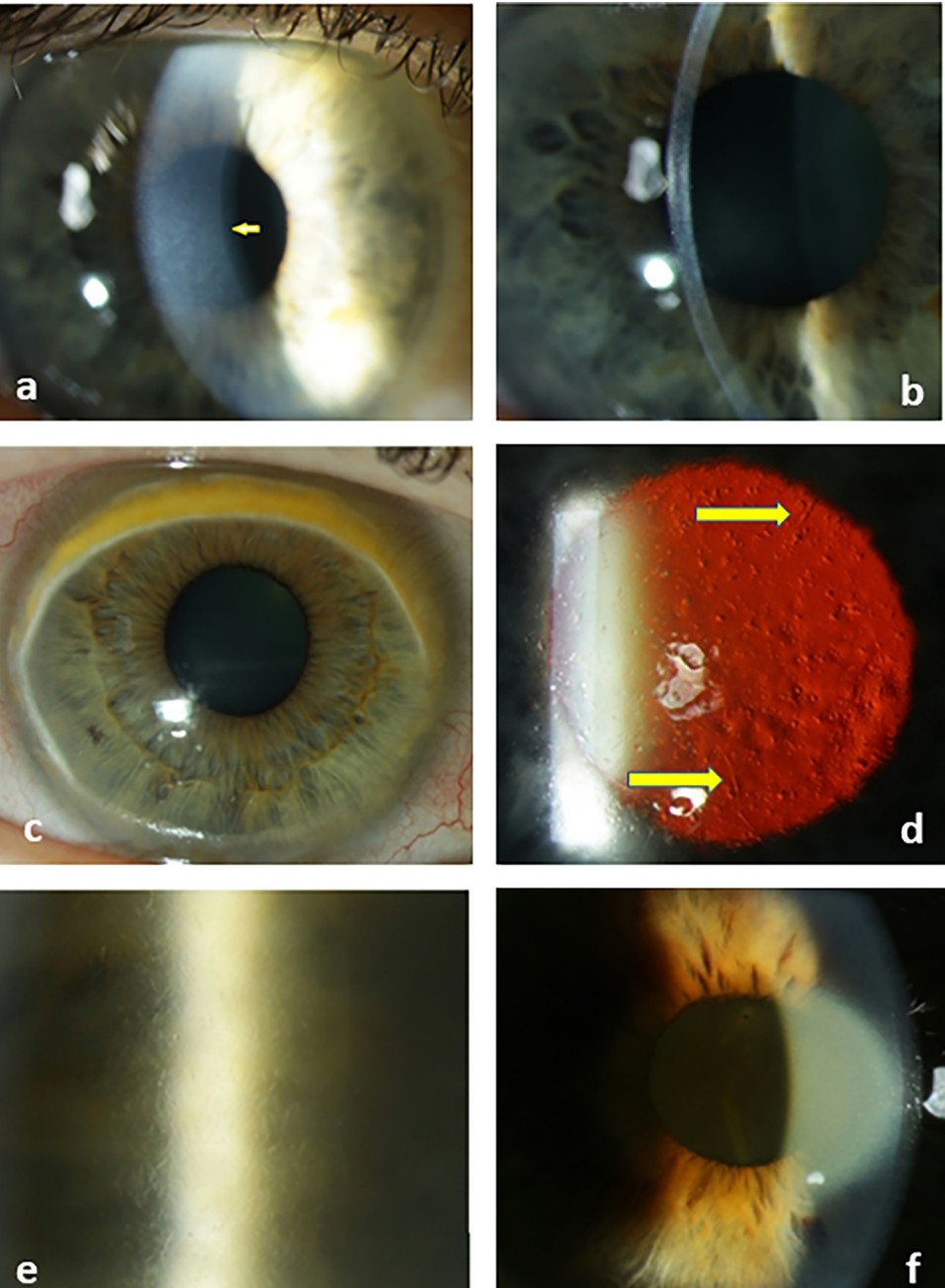

**Fig 1.** Stromal flake paraproteinemic keratopathy (PPK) in diffuse (a) and slit-shaped (b) light beam; c peripheral superficial band-like PPK; d lattice-like PPK in direct retro illumination (yellow arrows point the lattice lines); e stromal punctiform crystalline-like deposits; f central golden-brown discoloration of the pre-Descemet layer.

The median level of M-protein (p = 0.59), IgA (p = 0.53), IgG (p = 0.79) and IgM (p = 0.59) did not differ significantly between the patients with and without PPK. The median level of the FLC κ in serum of patients with kappa-restricted plasma cell dyscrasia was 209 mg/l ($25^{th}$ percentile 23.2 mg/l, $75^{th}$ percentile 631 mg/l) in patients with PPK compared to 38.1 mg/l ($25^{th}$ percentile 9.63 mg/l, $75^{th}$ percentile 83.0 mg/l) in patients without PPK (p = 0.18, FLC κ reference range 3.3–19.4 mg/l). The median level of the FLC λ in serum of patients with lambda-

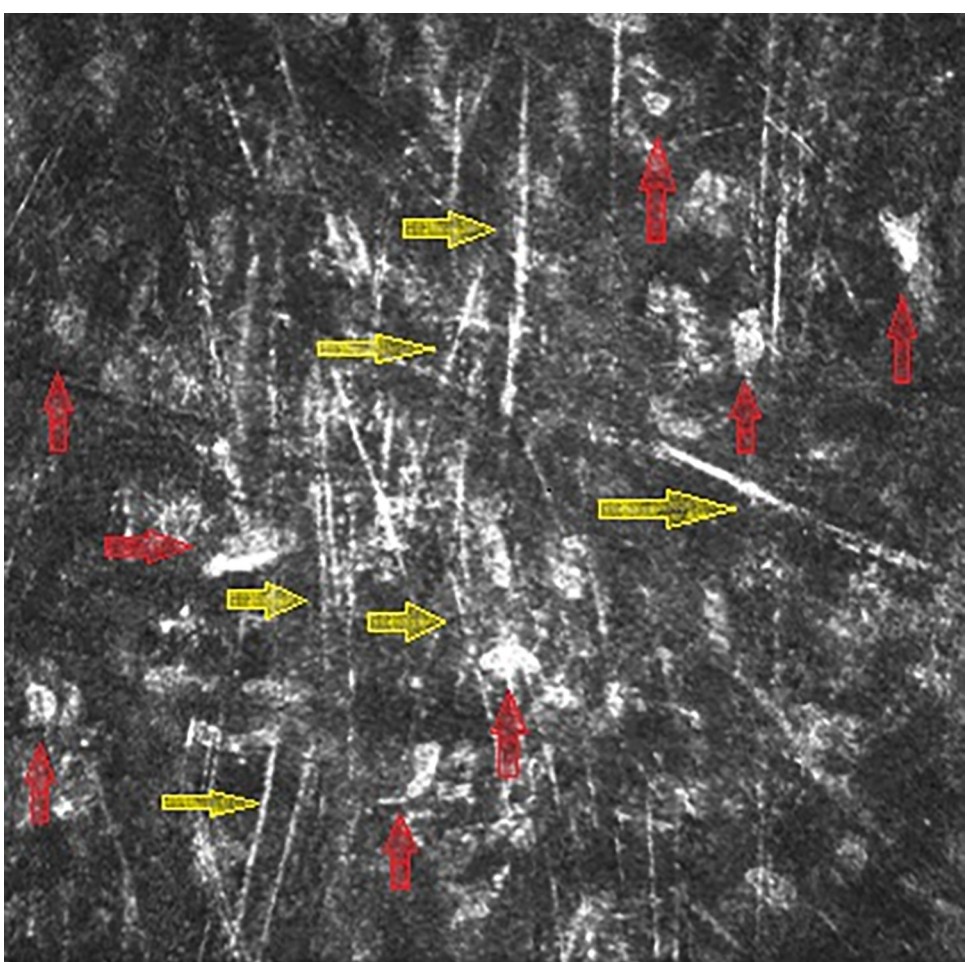

**Fig 2. In vivo confocal laser scanning microscopy: Extracellular hyperreflective needle-like deposits at the level of 284 μm in patient with stromal flake-like PPK associated with multiply myeloma of type IgG kappa.** Red arrows point keratocytes; yellow arrows point needle-like paraprotein deposits.

restricted plasma cell dyscrasia was 17.8 mg/l (25th percentile 13 mg/l, 75th percentile 33.9 mg/l) in patients with PPK compared to 418 mg/l (25th percentile 38 mg/l, 75th percentile 1950 mg/l) in patients without PPK (p = 0.02, FLC λ reference range 5.7–26.3 mg/l).

**Table 3. Comparison of hematological and ophthalmological findings between patients with PPK and patients without PPK.**

| Parameter (range) | PPK | | | non-PPK | | | |
|---|---|---|---|---|---|---|---|
| | 25th percentile | median | 75th percentile | 25th percentile | median | 75th percentile | p value |
| **FLC κ*** (3.3–19.4 mg/l) | 23.2 mg/l | 209 mg/l | 631 mg/l | 9.63 mg/l | 38.1 mg/l | 83.0 mg/l | 0.18 |
| **FLC λ**** (5.7–26.3 mg/l) | 13 mg/l | 17.8 mg/l | 33.9 mg/l | 38 mg/l | 418 mg/l | 1950 mg/l | 0.02 |
| **Parameter** | range | | median | range | | median | p value |
| **BCVA** (logMAR) | 0.8–0.0 | | 0.1 | 1.3–0.0 | | 0.0 | 0.10 |
| **corneal thickness** | 499–759 μm | | 558 μm | 485–613 μm | | 556 μm | 0.58 |
| **corneal densitometry** | 29–100 | | 36.4 | 26.8–60.6 | | 35.6 | 0.15 |

PPK–paraproteinemic keratopathy, FLC–free light-chain, K–type kappa, λ–type lambda, BCVA–best corrected visual acuity.

* in serum of patients with kappa-restricted plasma cell dyscrasia

** in serum of patients with lambda-restricted plasma cell dyscrasia

**Table 4. Serological types of the monoclonal gammopathy and morphological types of the paraproteinemic keratopathy (PPK) in patients with PPK.**

| PPK type | IgG κ n = | IgA κ n = | LC κ n = | LC λ n = | IgG λ n = | IgA λ n = |
|---|---|---|---|---|---|---|
| flake-like | 6 | 2 | | 2 | 1 | 3 |
| band-like | | | 1 | | | 1 |
| lattice-like | | | 1 | | | |
| golden-brown discoloration* | | | | | | 1 |
| crystalline-like | 1 | | | | | |

K–type kappa, λ–type lambda, Ig–Immunoglobulin, LC–light-chain.

* biclonal IgG and IgA lambda type MM

IgGκ is the most common serological type in patients with MG. Therefore, we performed a comparison of IgG and FLC κ values among the subgroup with MG type IgGκ using the Mann-Whitney-U test. This showed increased FLC κ in patients with PPK compared to patients without PPK (p = 0.03) but no significant difference in IgG (p = 0.91) in patients with PPK compared to patients without PPK.

## Discussion

To the best of our knowledge, this is the first prospective study evaluating corneal involvement in a large cohort of patients suffering from monoclonal gammopathy. Our primary aim was to evaluate prevalence of PPK in a population of patients suffering from monoclonal gammopathy. We found bilateral corneal opacities of heterogeneous severity and morphology in almost one third of the patients with the primary diagnosis of MG without any known pre-existing corneal pathology. This result confirmed our hypothesis that PPK is an underdiagnosed ocular condition and its prevalence among MG patients is higher than expected.

Our secondary aim was to evaluate whether the type and severity of MG determines the presence, morphology, and severity of the PPK. We hypothesized that severity and morphology of PPK correlate with serum paraprotein level and type of the underlying MG. PPK was found in 42% of precancerous MGUS patients compared to 26% of MM patients. Furthermore, the median levels of M-protein, IgG, IgA and IgM did not differ significantly between the patients with and without PPK. We did not observe a higher proportion of PPK in patients with MM in comparison to MGUS although MM is associated with end organ damage. This would suggest a lack of correlation between ocular findings and a progress of the hematological disease. Therefore, we did not confirm our second hypothesis.

The most common form of PPK was a bilateral diffuse stromal clouding (stromal flake-like PPK) observed in almost 74% of patients with PPK. The needle-like extracellular structures compatible with the paraprotein were visualized in the corneal stroma using IVCM in all cases of slit-lamp-PPK suspicious cases [2, 17, 19–21]. The PPK was mostly discreet and did not relevantly reduce the visual acuity. However, in two patients, the deterioration of the visual acuity was so significant that the PKs were needed to restore their visual function. We can certainly use the term MGOS [16] for these cases and would strongly argue in favour of an individualized approach to decide about MG-directed haematological treatment to prevent disabling ophthalmological pathology.

The prevalence of PPK has been described once, so far. Bourne et al. found crystalline keratopathy in only 1 of 100 patients with a confirmed diagnosis of MM and concluded that a slit lamp examination was not relevant for screening [1, 6]. The present study indicates a prevalence of PPK in MG (MGUS, SMM or MM) of 29%. This is much higher than previously assumed. However, most of the case reports in the literature portrayed the "gut visible"

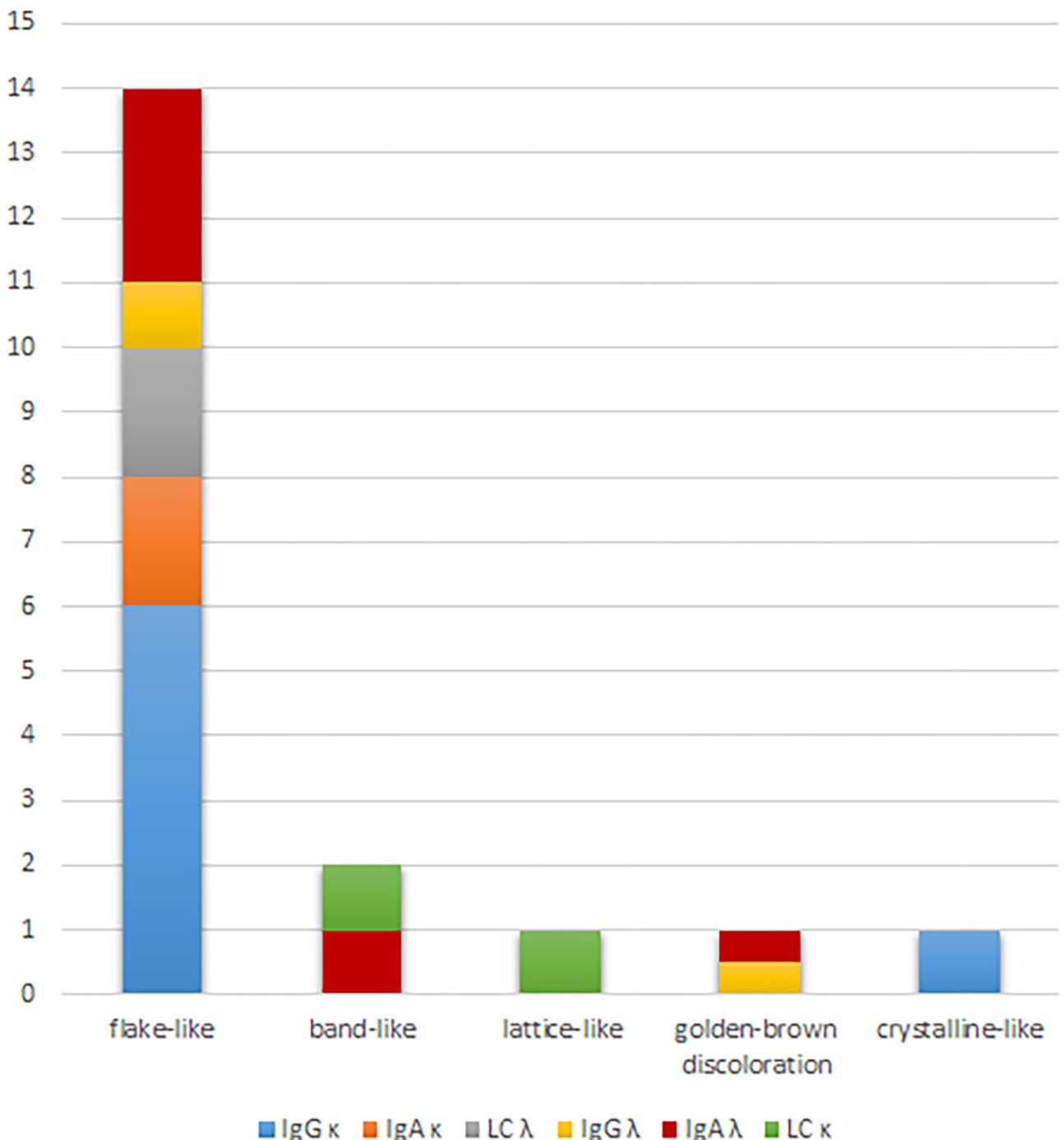

**Fig 3. Presentation of the different types of paraproteinemic keratopathy (PPK) with their case number and serological type.** X-axis: morphologic types of PPK; Y-axis: number of patients representing each type of PPK and the respective monoclonal paraprotein. K–kappa, λ–lambda, Ig–Immunoglobulin, LC–light-chain.

crystalline-like PPK. This is probably since the "non-crystalline" PPK types were unknown and subsequently overlooked or misdiagnosed. The most common form of PPK in the present study was a discreet stromal haze, which can easily be overlooked. The prevalence of crystalline keratopathy in our study was 2% (1 of 45 patients) and, hence, comparable with the prevalence found by Bourne at al.

Interestingly, the needle-like—crystalline—structures compatible with paraprotein were observed using IVCM in all our PPK cases independently from their morphological features observed in the biomicroscopy. We speculate that the non-crystalline types of PPK diagnosed on the slit lamp are crystalline on the level of the confocal microscopy. The classification in crystalline and non-crystalline deposits seems to be merely a "matter of magnification". Furthermore, using IVCM we observed hyperreflective point- and/or needle-like deposits in the deep stroma of patients without any PPK-like opacities visible on slit lamp. Future clinical trial with a healthy control group should answer the question if IVCM could detect the "pre-slit-lamp" PPK forms.

A relationship between the corneal opacity pattern and the serological findings in MG has not been investigated in a larger population so far. Lisch et al. evaluated 11 cases of PPK and considered several cases in the literature, including 13 cases of crystalline-like PPK, 6 cases of lattice-like PPK and 12 cases of pre-Descemet golden-brown opacity in MGUS and hypercupremia and proposed a new classification of PPK based on its morphology [1].

The majority of crystalline and non-crystalline PPKs is caused by the accumulation of IgG and LC κ [1, 22]. This is likely because about 70% of the MG cases are caused by the monoclonal proliferation of the IgG-restricted plasma cells [23]. Furthermore, the monoclonal proliferation of LC κ-restricted plasma cells occurs more frequently than of their LC λ-restricted counterparts. In our study, patients with plasma cell dyscrasia had both crystalline and non-crystalline PPK. On the other hand, patients with PPK sometimes had a high serum light-chain level (κ or λ) and sometimes a slightly elevated light-chain level, so that we could not directly speak of an effect of the size or charge of the light-chain in the diffusion properties through ocular structures and predilection sites of deposition in the cornea.

Singh et al. examined corneal transplants with recurrent PPK in MGUS of type IgG κ using electron microscopy and found the immunostaining for LC κ strongly positive [24]. Immuno-fluorescence and immunohistochemistry showed for the first time a link between LC κ and crystalline corneal deposits [21, 25, 26]. Furthermore, the present study revealed increased median serum FLC κ values in patients with kappa-restricted plasma cell dyscrasia and PPK compared to the patients with kappa-restricted plasma cell dyscrasia without corneal involvement. This difference is not statistically significant, admittedly the small sample size and rarity of the disease represent the main obstacles for meaningful statistical analyzes. This might not rule out a relationship between the absolute amount of increased FLC κ level and the probability of corneal deposits. IgG κ is also the most reported immunoglobulin fraction associated with corneal deposition [1]. Furthermore, corneal involvement in MG has been attributed to two factors in the literature: gammopathy of type IgG κ and long-term disease [1, 20, 27]. In the present study, there was no statistically proven association between the presence of MG type IgG κ and PPK, but within this subgroup of patients the level of FLC κ (by similar IgG levels) was significantly higher (p = 0.03) in patients with PPK compared to patients without PPK. On the contrary, the comparison of FLC λ among the patients with lambda-restricted plasma cell dyscrasia showed increased FLC λ in patients without PPK compared to patients with PPK (p = 0.02). This finding might be a chance finding. Furthermore, we did not observe any association between the median levels of M-protein, IgA, IgG and IgM and the presence of PPK or a higher proportion of PPK in patients with MM in comparison to MGUS although MM is associated with end organ damage. This would suggest a lack of correlation between

ocular findings and a progress of the hematological disease. Hematologists are interested in the size of the B-cell clone and the potential aggressiveness of a monoclonal component such as MM, Waldenström macroglobulinemia and MGUS. However, even a small clone, as occurs in MGUS, can produce a deleterious monoclonal protein that can be responsible for dangerous systemic organ damage, which can have a major impact on the quality of life, so that the term dangerous small B-cell clones has been increasingly required [28]. The aggregation of monoclonal proteins in the cornea can be a sign of possible further organ damage and timely treatment could prevent further organ damage.

With regard to the stromal crystalline-like PPK reported in the literature, the serological findings included in most cases IgG κ type [1, 3, 13, 23, 25, 29–42], in one case IgA κ type [43] and in two cases IgG λ type [13, 38]. In the present study, the patient with the stromal punctiform crystalline-like PPK had an SMM of type IgG κ.

In the present study, the flake-like PPK was identified in 6 patients with MGUS or MM of type IgG κ, 3 patients with SMM or MM of type IgA λ, 2 patients with MGUS or MM of type IgA κ, 2 patients with MM of type LC λ and a patient diagnosed with MGUS of type IgG λ. This shows the flake-like PPK accompanying different serological types. The stromal lattice-like PPK has almost always been described in the literature in connection with IgG κ [1, 5, 44–47]. In one case it was based exclusively on IgG [48] and in one case IgG κ was combined with IgM κ [44]. In the present study, the patient with a lattice-like PPK had a MGUS of type LC κ, which indicates an interrelation between LC κ and the lattice-like PPK.

The pre-Descemet golden-brown discoloration with hypercupremia was reported in the literature in 5 cases with IgG κ [49–53], in 5 cases with IgG λ [54–57] and in one case with IgG [58]. In the present study, the patient's serological findings indicate a biclonal MM of type IgG λ and IgA λ. This PPK-pattern seems to be related to IgG and hypercupremia.

The peripheral band-like PPK has so far only been shown in connection with IgG κ in a few reports in the literature [24, 59]. In the present study, the patients with the peripheral band-like PPK have a SMM of type IgA λ and a SMM of type LC κ. This form seems to occur less frequently and with different serological types.

Enders et al. compared the corneal densitometry between 5 patients with MG and 13 patients without MG in 4 different corneal layers without specific information on corneal involvement in MG patients and found that this was significantly higher in the anterior corneal layer in patients with MG than in patients without MG [60]. Likewise, Ichii et al. found that the corneal transparency in 30 MG patients without crystalline corneal deposits was significantly impaired compared to an age-adjusted control group [61]. In our study, we compared the densitometry of the central cornea between two groups of MG patients (with and without corneal opacity) and found no significant difference. This can be attributed to a high proportion of patients with discreete corneal opacity and to the fact that the densitometry was measured centrally in our study and the corneal opacities were often more pronounced peripherally.

Even though we collected the largest cohort of MG-patients with the corneal involvement, the relatively low number of cases is a limitation of our study. In Germany, the private hematologists rather than university medical centers take care of the most patients with MGUS or SMM. Hence, these patients visit the Hematological Department of the University Medical Center not so often. On the other hand, the patients suffering from MM are often seriously ill, so that an ophthalmological examination is not possible in many cases. In addition, according to our study protocol, the evaluation had to be carried out before the initiation of the systemic therapy. The future trials should recruit the patients also in the hematological private practices and not only at the university centers due to the low number of the "hematologically unremarkable" MGUS patients in the letter ones. We show trends, which must be confirmed in

further multicenter trials and recommend setting up an international registry for PPK patients. A further limitation of the study was an absence of a control group. This should be improved in further studies in order to rule out possible bias by the notification to the myeloma diagnosis.

In conclusion, PPK mostly occurs in form of a fine corneal stromal opacity and does not usually affect the visual acuity. However, advanced corneal opacification may even require corneal transplantation in order to restore the visual function. We recommend using the term MGOS for such patients as proposed earlier [16]. Since even minimal corneal findings can be a manifestation of a precancerous stage or of a systemic life-threatening disease, a hematological workup should be considered in all cases of bilateral corneal opacities of unknown entity. Furthermore, we strongly recommend an ophthalmological workup of all patients suffering from monoclonal gammopathy in order to rule out a corneal or retinal involvement.

## Supporting information

**S1 Checklist. TREND checklist.**
(PDF)

**S1 Protocol. Study protocol in German.**
(DOC)

**S2 Protocol. Study protocol in English.**
(DOC)

**S1 File. Ethics application in German.**
(DOCX)

**S2 File. Ethics application in English (translated parts highlighted yellow).**
(DOCX)

**S3 File. Distributions in boxplots.**
(DOCX)

**S1 Dataset. Minimal data set of the study.**
(XLSX)

## Acknowledgments

We thank all study participants for their willingness to provide data for this research project and we are indebted to all coworkers for their enthusiastic commitment.

## Author Contributions

**Conceptualization:** Mohammad Al Hariri, Markus Munder, Walter Lisch, Alexander Desuki, Adrian Gericke, Joanna Wasielica-Poslednik.

**Data curation:** Markus Munder, Alexander K. Schuster.

**Formal analysis:** Mohammad Al Hariri, Markus Munder, Alexander K. Schuster, Joanna Wasielica-Poslednik.

**Investigation:** Mohammad Al Hariri, Markus Munder, Eva-Marie Fehr, Björn Jacobi, Alexander Desuki, Andreas Kreft, Joanna Wasielica-Poslednik.

**Methodology:** Mohammad Al Hariri, Alexander K. Schuster, Alexander Desuki, Adrian Gericke, Joanna Wasielica-Poslednik.

**Project administration:** Mohammad Al Hariri, Joanna Wasielica-Poslednik.

**Resources:** Norbert Pfeiffer.

**Software:** Alexander K. Schuster.

**Supervision:** Markus Munder, Norbert Pfeiffer, Joanna Wasielica-Poslednik.

**Writing – original draft:** Mohammad Al Hariri, Joanna Wasielica-Poslednik.

**Writing – review & editing:** Markus Munder, Walter Lisch, Alexander K. Schuster, Eva-Marie Fehr, Björn Jacobi, Alexander Desuki, Andreas Kreft, Adrian Gericke, Norbert Pfeiffer.

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
