## [Decision Letter · Decision Letter 0]

13 Oct 2021

PONE-D-21-09281Correlation between corneal and hematological findings in monoclonal gammopathyPLOS ONE

Dear Dr. Wasielica-Poslednik,

Thank you for submitting your manuscript to PLOS ONE. After careful consideration, we feel that it has merit but does not fully meet PLOS ONE’s publication criteria as it currently stands. Therefore, we invite you to submit a revised version of the manuscript that addresses the points raised during the review process.

I would like to sincerely apologise for the delay you have incurred with your submission. It has been exceptionally difficult to secure reviewers to evaluate your study. We have now received four completed reviews; their comments are available below. The reviewers have raised significant scientific concerns about the study, in particular about the study design and methodology, that need to be addressed in a revision.

Please revise the manuscript to address all the reviewer's comments in a point-by-point response in order to ensure it is meeting the journal's publication criteria. Please note that the revised manuscript will need to undergo further review, we thus cannot at this point anticipate the outcome of the evaluation process.

We look forward to receiving your revised manuscript.

Kind regards,

Miquel Vall-llosera Camps

Senior Editor

PLOS ONE

Journal Requirements:

2. Thank you for submitting your clinical trial to PLOS ONE and for providing the name of the registry and the registration number. The information in the registry entry suggests that your trial was registered after patient recruitment began. PLOS ONE strongly encourages authors to register all trials before recruiting the first participant in a study.

1) your reasons for your delay in registering this study (after enrolment of participants started);

2) confirmation that all related trials are registered by stating: “The authors confirm that all ongoing and related trials for this drug/intervention are registered”

Reviewers' comments:

Reviewer's Responses to Questions

**Comments to the Author**

1. Is the manuscript technically sound, and do the data support the conclusions?

Reviewer #1: Yes

Reviewer #2: Partly

Reviewer #3: Partly

Reviewer #4: Partly

2. Has the statistical analysis been performed appropriately and rigorously? 

Reviewer #1: Yes

Reviewer #2: Yes

Reviewer #3: I Don't Know

Reviewer #4: No

3. Have the authors made all data underlying the findings in their manuscript fully available?

Reviewer #1: No

Reviewer #2: Yes

Reviewer #3: Yes

Reviewer #4: Yes

4. Is the manuscript presented in an intelligible fashion and written in standard English?

Reviewer #1: Yes

Reviewer #2: Yes

Reviewer #3: No

Reviewer #4: Yes

5. Review Comments to the Author

Reviewer #1: This paper is an interesting evaluation of ophthalmic findings in patients with MGUS or myeloma.

As this was a prospective study, but not randomised, the CONSORT diagram (for randomised trials) should be replaced with a flowchart lacking the second column of boxes. It would appear that all cases of MG in the centre were in this study - is this true? Does this represent all patients seen at your centre? If so, Figure 1 is largely unnecessary and can be replaced by a statement saying there is complete ascertainment of all cases.

Why was 51 considered enough? How was the sample size decided upon here? Was it a particular size or time period in the prospective design of the study?

There needs to be a table of patient characteristics, and an understanding if these are newly diagnosed patients, or at what stage in their disease are they measured? One assumes that this is a single measurement taken at or soon after diagnosis - but this is not made explicit.

One assumes the worse eye for each patient is used here to make the data independent - is this right?

Figure 3 would be clearer as a table with a test for association.

The Mann Whitney U test does not compare medians, but rather distributions. Pleas therefore change the statements about medians. Data need to be given in addition to just p-values. To what extent is a non-significant result sufficient to rule out a meaningful difference?

Reviewer #2: Please find my comments as an attachment file. --------------------------------------------------------------------------------------------------------------------------------------------------------------------------------------------------------------------------

Reviewer #3: In this work, Hariri and colleagues characterized monoclonal gammopathy (MG) patients with paraproteinemic keratopathy (PPK). I provide the following comments to assist the authors in revising and improving their manuscript and study:

Major comments:

1) They includes patients who were diagnosed with PKK prior to MG diagnosis. Analyses with patients from group A and B could reduce the accuracy of the study.

2) Results with slit lamp examination are inconsistent with previous studies. It might be possible that the data could be biased by the notification of the myeloma diagnosis. They need to show how to avoid the bias. Examination results from age-matched controls would be useful.

3) The increase in density measured with Scheimpflug tomography is expected based on the data from slit lamp examination or IVCM. Why did the density show no differences between patients with/without PPK? The density might increase in MM patients, regardless of the complication of PPK (Int J Hematol. 2019 Oct;110(4):500-505. Cornea. 2017 Apr;36(4):470-475.)?

4) Regarding case presentations, they did not mention whether PPK was induced by the MGUS or LPL (Line 222-231). LPL cells usually express IgM. They need to confirm that the tumor cells secret M-protein (IgG), or explain how LPL treatment could improve MGUS status.

5) As the authors mentioned in Method, the level of M-protein is one of diagnostic criteria for MGUS/SMM/MM. The level should be compared among patients with the same disease. The content about the possible correlation between the type and level of M-protein and PPK in discussion part is not convincing.

6) Certain types of immunoglobulin produced by a B cell lineage clone can be deposited at intra- or extracellular locations, forming amyloid fbrils, amorphous punctate aggregates, microtubules and crystals in several organs such as the kidneys, heart, skin and neurons. Among these organ damages, the disease concept of MGRS has been approved, because serious renal dysfunction is life-threatening. They should clarify the reason why the concept of “MGOS” is required.

Minor comments:

1) Figure1 is not needed.

2) It is better to show patient characteristics in a table.

3) Most of the results about hematological evaluation are missing.

4) lymphoplasmocytoid immunocytoma (line 223) ⇒ lymphoplasmacytoid immunocytoma

5) BJP type can be diagnosed by urine sample examination.

Reviewer #4: The authors report corneal findings in patients diagnosed with monoclonal gammopathy with different disease severities such as MGUS, smoldering MM, and MM investigating the prevalence of paraproteinemic keratopathy (PPK) and involvement of specific proteins. They included 51 patients from hematology clinics with monoclonal gammopathy and evaluated them for corneal findings per their study protocol. They found that 19/51 patients had PPK, indicating 37% prevalence, which is higher than previously reported. They also found that FLC kappa was higher but not statistically significant in patients with PPK vs. without PPK, and FLC lambda was significantly lower in patients with PPK vs. without PPK. The study and results are relevant. They need to clarify some points in their methods and revise results and discussion accordingly.

Overall comments:

1. The authors state some correlations or associations they found or did not find throughout the text; however, no analysis was performed to check for correlations or associations. They need to revise wording throughout the manuscript, including the title.

2. Methods need some clarification about the recruitment and application of study procedures. Patients were probably recruited from the hematology department based on the inclusion/exclusion criteria, but it is unclear. It should be clarified where the study procedures were conducted and in which order. Hematological and ophthalmological evaluation subheadings require further detail in this regard. After the consent was obtained, what procedures were performed in which order, whether the blood test was performed on the same day or within a specific interval with the ophthalmologic evaluation.

3. Recommending having three study groups based on the classification of the hematologic disease. Then, they can have two subgroups based on the presence or absence of PPK to compare hematologic measures. It does not make sense to have groups A and B based on PPK was known or unknown as it does not help address any question.

Specific comments:

1. The abstract purpose may need to be rephrased to determine the prevalence of PPK among patients with MG.

2. Abstract methods or results would consider adding n for each group as MGUS, SMM, and MM.

3. According to this study, the prevalence of PPK in patients with MG was 37% (19/51), not 29%.

4. Lines 120-130: Further details needed regarding hematologic measures, when and where the laboratory test was performed? At a research lab or a certified clinical lab, where exactly? What were the unit measures?

5. Lines 134-140: Further details are needed regarding ophthalmologic evaluations. In what order were the procedures performed? How was the VA reported? Was Snellen converted to logMAR for analysis or not? Slit-lamp biomicroscopy and fundoscopy were explicitly performed to assess what details? What were parameters from tomography and OCT assessed?

6. CONSORT flowchart in its current version does not seem helpful to give sufficient detail for this study. As this was a clinical trial, the CONSORT flowchart may not need to be used here. A revised flowchart can be helpful, though, to clarify the points mentioned above.

7. Lines 156-158: A table with demographic and clinical characteristics of study participants would be helpful.

8. Lines 159 and 162: It is confusing what these study groups refer to. How are these 6 patients different from 45 patients included in group B? If there were study groups, they should be defined in the Methods section first. If any subgroup analysis was planned, again, it should be described in the Methods section first. Authors may consider having 3 study groups based on the hematologic diagnosis as MGUS, SMM, and MM. See above for further details.

9. Lines 163-165: PPK in 19/51, 37% prevalence, among all participants is as important. A second table displaying the results between 163-170 (and maybe other results) according to the hematologic condition would be helpful.

10. Figure 3 does not show a correlation. Legend needs revision and a better description of what was displayed.

11. Lines 193-201: Another table including demographic and clinical characteristics of patients according to PPK status would be helpful.

12. Line 216: Were all these patients followed up or determined whether they needed PK based on the initial assessment, or only these two?

13. Lines 219-231: How long did the study require to follow up of patients? What parameters were checked during the follow-up? Did they find any of the participants with no PPK at baseline developed PPK later?

14. Line 263-265: Consider including the reference paper that proposed MGOS terminology.

15. Line 339-340: “We recommend using the term MGOS for such patients” should be followed by “as proposed earlier” and relevant reference should be included.

6. PLOS authors have the option to publish the peer review history of their article (what does this mean?). If published, this will include your full peer review and any attached files.

Reviewer #1: No

Reviewer #2: **Yes: **Linus Alexander Völker

Reviewer #3: No

Reviewer #4: No

---

## [Author Response · Author response to Decision Letter 0]

12 Dec 2021

PONE-D-21-09281

Correlation between corneal and hematological findings in monoclonal gammopathy

Dear Editor,

we thank you and the reviewers for the valuable comments and suggestions and hope to have improved our manuscript now. Below you will find our responses to the editor´s and reviewers´ comments.

With kindest regards

Joanna Wasielica-Poslednik

Our answer: We formatted our manuscript according to the PLOS ONE requirements.

2. Thank you for submitting your clinical trial to PLOS ONE and for providing the name of the registry and the registration number. The information in the registry entry suggests that your trial was registered after patient recruitment began. PLOS ONE strongly encourages authors to register all trials before recruiting the first participant in a study.

1) your reasons for your delay in registering this study (after enrolment of participants started);

Our answer: The study was submitted to the ethics committee in April 2016. The date of the positive vote / approval of the ethics committee of the State Medical Association of Rhineland-Palatinate was 22.09.2016. All participants gave their consent and were included in the study after the ethics committee had given the positive vote

The date of registration in the DRKS (German Clinical Trial Register: DRKS00023893) was 11.01.2021. According to the law, study participants in Germany may be recruited after the positive vote has been given. A registration in the DRKS is not required. There was no special reason for this delay. We will register our future studies earlier.

2) confirmation that all related trials are registered by stating: “The authors confirm that all ongoing and related trials for this drug/intervention are registered”

Our answer: The authors confirm that all ongoing and related trials for this drug/intervention are registered.

Our answer: We added our minimal data set as a Supporting Information file (S6) 

Responses to reviewers’ comments

We thank all reviewers for their valuable comments and suggestions and hope to have answered all their questions. 

Reviewer #1: This paper is an interesting evaluation of ophthalmic findings in patients with MGUS or myeloma.

As this was a prospective study, but not randomised, the CONSORT diagram (for randomised trials) should be replaced with a flowchart lacking the second column of boxes. It would appear that all cases of MG in the centre were in this study - is this true? Does this represent all patients seen at your centre? If so, Figure 1 is largely unnecessary and can be replaced by a statement saying there is complete ascertainment of all cases.

Our answer: We agree that the CONSORT diagram is not the proper one for this non-randomized trial. However, the CONSORT diagram is necessary according to the journal´s requirements. Hence, we followed your suggestion and modified (simplified) it. 

Why was 51 considered enough? How was the sample size decided upon here? Was it a particular size or time period in the prospective design of the study?

Our answer: The patient recruitment was limited in time. We aimed to recruit a maximum of consecutive patients suffering from MGUS, SMM and MM in a period of maximal 3 years. In Germany, the private hematologists take care of the most patients with MGUS or SMM. Hence, these patients are not so often at the Hematological Department of the University Medical Center. On the other hand, the patients suffering from MM are often seriously ill, so that an ophthalmological examination is not possible in many cases. In addition, according to our study protocol, the evaluation had to be carried out before the initiation of the systemic therapy. All these circumstances made the recruitment quite difficult. That is why we recruited “only” 51 patients in this quite a long period. We added this information in lines 119-120 (version with mark-ups) in the Methods section as well as in lines 438-446 in the Discussion.

There needs to be a table of patient characteristics, and an understanding if these are newly diagnosed patients, or at what stage in their disease are they measured? One assumes that this is a single measurement taken at or soon after diagnosis - but this is not made explicit.

Our answer: We added table 1 with patients´ characteristics in the results section. 

The aim of the study was to examine the eyes of as many patients suffering from monoclonal gammopathy (MGUS, SMM, MM) as possible in order to find out the prevalence of the associated keratopathy (PPK). This was a case in 45 patients, who received the hematological diagnosis of MG at the Hematological Department and received the ophthalmological evaluation within 3 months afterwards. However, there were additional 6 patients who received the ophthalmological diagnosis of PPK before the hematological diagnosis. We decided to include these 6 patients to our analysis, but not to the prevalence assessment. We added this information as well as the period of the measurements in lines 121-134 (version with mark-ups) in the Methods section. 

One assumes the worse eye for each patient is used here to make the data independent - is this right?

Our answer: The corneal changes were mostly the same in both eyes and the testing was done by person and not by eye.

Figure 3 would be clearer as a table with a test for association.

Our answer: Thank you very much for this valuable suggestion. We added table 2 in the Results section. We also performed an association test (Spearman's correlation) and found no correlation, so we revise the wording throughout the manuscript, including the title, as requested by Reviewer # 4. 

The Mann Whitney U test does not compare medians, but rather distributions. Please therefore change the statements about medians. Data need to be given in addition to just p-values. To what extent is a non-significant result sufficient to rule out a meaningful difference?

Our answer: We ran the Mann Whitney U test to compare the different distributions. We would like to keep the information on the medians as additional information and provide the distributions in a supplementary file (S7).

We performed the test on small groups and recommend applying this test to larger groups in order to confirm or exclude a meaningful difference.

Reviewer #2: Please find my comments as an attachment file. 

Our answer: Thank you for your comments and suggestions. We copied your comments and answered them in this file. 

The authors report the result of a prospective observational cohort of 51 patients with MGUS, SMM, or MM describing bilateral corneal opacities in an unexpectedly high percentage of patients. Additionally, they describe the clinical courses and managements of two patients with severe ocular affections. The manuscript is well-written, concise and summarizes the findings in a valid manner. 

Our answer: Thank you very much for your kind assessment.

However, before considering publication in PLOSone, I suggest some revisions to the manuscript. 

1. Throughout the manuscript, the 51 included patients are divided into patients with a prior hematologic diagnosis (i.e. 45 patients, “group B”) and without such a diagnosis (6 patients, “group A”). One may assume that information on the hematologic diagnosis of these patients has become available at a later point. Please clarify why the distinction between these two patient groups is necessary and provide information on their later hematologic diagnosis or remove the separation into two groups from the manuscript since it is not represented in the figures nor sufficiently discussed. 

Our answer: Thank you for this comment. The primary aim of the study was to find out the prevalence of paraproteinemic keratopathy in patients suffering from monoclonal gammopathy. The 45 patients from group B first received the hematological diagnosis of the monoclonal gammopathy and later on received an ophthalmological evaluation to find out if they have PPK. We assessed the prevalence of the paraproteinemic keratopathy based on the results from this group.

However, there were additional 6 patients (group A), who first received the ophthalmological diagnosis of the paraproteinemic keratopathy without any known hematological disorder. Actually, in group A a hematological workup was initiated on ophthalmologist´s request. 

We decided to include these 6 patients (group A) to our analysis, but not to the prevalence assessment. That is why we divided the cohort into two groups. We added this information in lines 121-130 (version with mark-ups) in the Methods section and hope to have clarified this point. We modified Fig. 1 as well.

Furthermore, the history of the patients from group A should be a kind message for the ophthalmologists to consider a hematological workup in case of any bilateral corneal opacity of unclear dignity.

2. In line 62-63, the “latest classification of PPK” is mentioned and cited. Please expand to provide more information on that classification to the reader. 

Our answer: Thank you. We added some more information and reference no. 5 in line 66.

3. In the introduction, it should be mentioned that MG is not always caused by an underlying plasmacell disorder but can be related to any form of b-cell proliferative disease (Lymphome, CLL, etc.). This is particularly important since the patient case presented later in the manuscript features a lymphoplasmocytoid immunocytoma. 

Our answer: Thank you for this recommendation. We added it in lines 71-72.

As a little clarification – the patient with lymphoplasmacytic lymphoma also had MGUS type IgG kappa.

4. Lymphoplasmocytoid immunocytoma should be changed to lymphoplasmacytic lymphoma.

Our answer: Done

5. Line 75, reference 9 is incorrect and should be changed to [ https://doi.org/10.1159/000068251 ]

Our answer: Done

6. In line 84, current hematologic guidelines should be cited in addition to reference 17

Our answer: Thank you. We added it as ref. 18.

7. Line 89, correct to “A further goal.”

Our answer: Done

8. Line 102-103, it is inconclusive why 6 patients without a hematologic diagnosis solely based on the finding of PKK were included in the study considering the given inclusion criteria. Please clarify.

Our answer: In these 6 patients an ophthalmological evaluation aroused suspicion of a hematological disorder. The following hematological workup confirmed this suspicion in all cases. We included these patients in the study after the hematological diagnosis had been confirmed.

9. Line 103, “MG” is supposed to be “MM”? Please clarify the difference between MG and MGUS as used in the manuscript and make its use in the manuscript consistent.

Our answer: Thank you for this correction. We changed it to MM. 

10. The exclusion criterium of “non-measurable M-protein” appears detrimental to the study because it prevents patients to be included who have had a complete response to therapy or b-cell clones with unmeasurable yet harmful paraproteins (as is the case in various renal paraprotein deposition diseases). Please specify if patients were excluded based on previous therapies, which assays were used to quantify m-protein, and the rational for this exclusion criterium. 

Our answer: All patients who had already received therapy were excluded because the therapy could possibly cause a change in the corneal condition. Without a measurable M protein, the diagnosis of systemic MG cannot be confirmed; in such cases (as in various renal paraprotein deposition diseases), the patients were excluded.

11. Line 109, slimCRAB criteria should be used to define MGUS. 

Our answer: We added it.

12. The concept of “monoclonal gammopathy of renal significance” and “monoclonal gammopathy of clinical significance” should be briefly discussed in this paragraph. 

Our answer: Done in lines 148-151 (version with mark-ups)

13. Line 116, “special biomarkers of malignancy” should be specified. I suppose this refers to slimCRAB. Also, cast nephropathy is a myeloma-defining event and should be added. 

Our answer: Done

14. Lines 120-130, please make capitalization of parameters consistent.

Our answer: Done

15. Line 129, “immune fixation” should be “immunofixation”.

Our answer: Done 

16. Line 148, throughout the text, the correct use of “free light-chain” should be observed. 

Our answer: Thank you for the correction.

17. Line 150, information on software lacks vendor and country. 

Our answer: We added this information.

18. Flow chart, it seems peculiar that 51 consecutive patients could be included without a single patient not meeting the inclusion criteria indicating screening failure. It appears that patients were included based on their referral from the Department of Hematology, and that screening took place before that. This should be clarified in the text and, if applicable, added to the flow chart. At this point, it is unclear how patients were included in the study given these facts and the fact that 6 patients were diagnosed as primary PPK-patients without preceding hematologic diagnosis.

Our answer: In fact, the most study participants were examined hematologically independently of the study and once the patients met the hematological inclusion criteria, they were recruited into the study. 

We agree that the CONSORT diagram (Fig. 1) is not the proper one for this non randomized trial. However, the CONSORT diagram is necessary according to the journal´s requirements. Hence, we followed your suggestion and modified (simplified) it. 

Regarding the 6 patients please see our answer to the first question. 

19. The flow chart is lacking N numbers in some lines (i.e. “0”?). Please correct.

Our answer: We modified the flow chart.

20. A tabular view of the patients’ baseline characteristics should be given as an additional table. 

Our answer: We added table 1 in the results section.

21. A tabular view of the patients diagnoses and ophthalmologic findings should be given as an additional table (e.g. x-axis hematologic diagnosis, y-axis ophthalmologic diagnosis) to present data described in the results section in a more accessible way. This should replace the rather confusion figure 3. Moreover, summarizing patients according to kappa or lambda-paraprotein expression may help extract more information from this cohort.

Our answer: We added table 2 according to your suggestion.

22. The resolution of the images in figure 2-5 appears rather low. At this quality I cannot assess whether descriptions and arrows are accurate. Resolution should be increased for final publication. 

Our answer: We provided PLOS-ONE with images of a very good resolution. The quality of the images will be certainly better in the final version. 

23. Figure 5 d – it should be clarified when the in vivo confocal scanning microscopy was done before or after penetrating keratoplasty and whether this represents an ex vivo specimen or was done before the penetrating keratoplasty. In this case, I suggest rearranging the panels to reflect the fact that panel d predates panel c in time. 

Our answer: The in-vivo laser scanning microscopy was performed before the penetrating keratoplasty – we added this information in the legend to figure 5. We decided not to rearrange the panel as the figures in the upper row present clinical findings (biomicroscpy) and the figures in the bottom row present microscopic findings of the same tissue (confocal microscopy preoperative and histology postoperative). 

24. Do the authors have any information on the subclass of immunoglobulin types? This would provide valuable information and could explain the lack of correlation between MG-type and m-protein levels and ocular affection in this study. The current understanding is that paraprotein depositions are mainly driven by biochemical properties and less by total MG-levels (e.g. proliferative glomerulonephritis with monoclonal immunoglobulin deposits – PGNMID shows a clear correlation to IgG3-kappa subtype). These concepts should be discussed in the manuscript and information on the subclass of Ig should be presented if available. 

Our answer: Thank you for this comment. We have information about the Ig type (IgG, IgA, IgM) but not about subclasses (such as IgG1 and IgG3).

25. Line 258, the authors should discuss whether MGOS with rapid deterioration of visual acuity (as described in the case vignette) generally warrants systemic cytotoxic therapy. 

Our answer: We would say yes. We discuss on this topic in the discussion in lines 341-346.

26. Lines 288-292, as mentioned above, the biochemical properties play an important role in the likeliness of paraprotein to form organized or non-organized deposits and the predilection site of deposition. The authors should expand on the biochemical properties and (mainly size-) differences kappa- and lambda-light-chains and whether size and charge may affect diffusion properties through ocular structures and predilection sites of deposition in the eye. 

Our answer: Thank you for this comment. We discussed it in lines 373-377 (version with mark-ups. We find your point very interesting. It could be a topic of the next study (e.g., electron microscopic evaluation of the explanted corneas) 

27. Line 299, in the statement about the statistical significance, the small sample size and rarity of the disease should be discussed as the main obstacle to meaningful statistical analysis rather than speculations. 

Our answer: Thank you for this comment. We added this comment in lines 384-386.

28. Line 312, “advancement” should be changed to “progress”. 

Our answer: Done

29. Line 313, the concept of “dangerous small B-cell clones” (https://doi.org/10.1182/blood-2006-03-001164) should be discussed in this context. 

Our answer: Thank you for this nice addition. We added it to our discussion in lines 399-406 and added as the reference no.29.

30. Line 335-336, the statement on multicenter trials is rather bold. I suggest recommending setting up patient registries which appears more practical for rare disease such as PKK. 

Our answer: Thank you for this recommendation. We added it in line 447.

31. Lines 337-338. The sentence (In conclusion,…) needs to be revised to meeting standard English criteria.

Our answer: Thank you for this comment. We go with it.

32. I encourage the authors to propose an ophthalmologic guidance for the work-up of patients with paraproteinemia: should all patients undergo ophthalmologic evaluation? If not, who should be screened and how often? Does every ocular involvement warrant a cytotoxic therapy? Screening for recurrence after cornea transplantation? Should every patient be transplanted?

Our answer: Thank you very much for this recommendation. 

We strongly recommend an ophthalmological examination of all patients with MG for corneal or retinal involvement. We consider systemic therapy necessary for corneal involvement when visual acuity is impaired and before PK. Nevertheless, a lifelong follow-up after PK is necessary to rule out recurrence. Not every patient needs a corneal transplantation, only those with a relevant visual impairment.

We added these recommendations at the end of the discussion.

33. Line 340-341, the statement that morphology and severity of PPK do not correlate with the stage of monoclonal gammopathy appears bold in view of the given data and should be toned done. Moreover, there are no official stages of monoclonal gammopathy, this term seems misleading and should be corrected. 

Our answer: Thank you for this comment. I removed this statement. 

Reviewer #3: In this work, Hariri and colleagues characterized monoclonal gammopathy (MG) patients with paraproteinemic keratopathy (PPK). I provide the following comments to assist the authors in revising and improving their manuscript and study:

Major comments:

1) They includes patients who were diagnosed with PKK prior to MG diagnosis. Analyses with patients from group A and B could reduce the accuracy of the study.

Our answer: 

Thank you for this comment. The primary aim of the study was to find out the prevalence of paraproteinemic keratopathy in patients suffering from monoclonal gammopathy. The 45 patients from group B first received the hematological diagnosis of the monoclonal gammopathy and later on received the ophthalmological evaluation. We assessed the prevalence of the paraproteinemic keratopathy based on the results from this group.

However, there were additional 6 patients (group A), who first received the ophthalmological diagnosis of the paraproteinemic keratopathy. At this time the hematological diagnosis wasn´t even known. Actually, in group A the hematological workup was initiated on ophthalmologist´s request. 

We decided to include these 6 patients (group A) to our analysis, but not to the prevalence assessment. That is why we divided the cohort into two groups. We added this information in lines 121-134 (version with mark-ups) in the Methods section and hope to have clarified this point. We modified Fig. 1 as well.

Furthermore, the history of the patients from group A should be a kind message for the ophthalmologists to consider a hematological work-up in case of bilateral corneal opacity of unclear dignity.

2) Results with slit lamp examination are inconsistent with previous studies. It might be possible that the data could be biased by the notification of the myeloma diagnosis. They need to show how to avoid the bias. Examination results from age-matched controls would be useful.

Our answer: All morphological types of PPK in our study match with so far the only official morphological classification of PPK proposed by Lisch et. al. It is true that the examiners were not blinded to the hematological diagnosis of the patients and this could potentially make them more attentive to discreet corneal findings. On the other hand the examiners are experienced cornea specialists, who see hundreds of patients every month. A bias in the slit lamp examination was ruled out by confocal microscopy. All patients with the slit-lamp-diagnosed-PPK revealed paraprotein deposits in the confocal microscopy. We aim to include a control group in further studies. We added this limitation in the discussion in lines 448-449.

3) The increase in density measured with Scheimpflug tomography is expected based on the data from slit lamp examination or IVCM. Why did the density show no differences between patients with/without PPK? The density might increase in MM patients, regardless of the complication of PPK (Int J Hematol. 2019 Oct;110(4):500-505. Cornea. 2017 Apr;36(4):470-475.)?

Our answer: Thank you for this comment. We actually expected that the corneal density in the MG patients with PPK would be higher than in the MG patients without PPK. This may be a result of a high proportion of patients with discreet corneal opacity and because the density in our study was measured centrally and the corneal opacities were often more pronounced peripherally. We discussed it in lines 427-436 and added both references.

4) Regarding case presentations, they did not mention whether PPK was induced by the MGUS or LPL (Line 222-231). LPL cells usually express IgM. They need to confirm that the tumor cells secret M-protein (IgG), or explain how LPL treatment could improve MGUS status. 

Our answer: No IgM paraproteinemia could be detected, although histomorphologically a clear-cut LPL was present. This is not completely unusual, since around 5% of LPL cases are associated with a non-IgM paraprotein (usually IgG). We added this information in the results section: “We did not find any evidence of plasma cells in the histological examination of the cornea. Kappa light chain staining showed focal positivity of an embedded amorphous material, also staining for IgG was positive in amorphous material between the connective tissue lamellae of the cornea. The Congo stain showed no evidence of amyloid.” Lines 302-306. Treatment with R-Bendamustin and Bortezomib-Dexamethasone is active both against a LPL clone as well as against a classical MGUS plasma cell clone. Until now, a stable remission was induced in order to avoid or delay a recurrence of the corneal opacity after the keratoplasty.

5) As the authors mentioned in Method, the level of M-protein is one of diagnostic criteria for MGUS/SMM/MM. The level should be compared among patients with the same disease. The content about the possible correlation between the type and level of M-protein and PPK in discussion part is not convincing.

Our answer: As the PPK can occur in all MG patients, one of the aims of the study was to answer the question of whether the level of M-protein increases the risk of corneal involvement, i.e. whether the risk of corneal involvement is higher in MM patients than in MGUS patients. The number of study participants in these subgroups is very different and sometimes too little to compare (e.g. SMM n = 5). 

The median level of M-protein did not differ significantly between the patients with and without PPK (p = 0.59) –we mentioned it in the line 263 in the results section.

6) Certain types of immunoglobulin produced by a B cell lineage clone can be deposited at intra- or extracellular locations, forming amyloid fibrils, amorphous punctate aggregates, microtubules and crystals in several organs such as the kidneys, heart, skin and neurons. Among these organ damages, the disease concept of MGRS has been approved, because serious renal dysfunction is life-threatening. They should clarify the reason why the concept of “MGOS” is required.

Our answer: Thank you for this comment. Corneal opacity in the context of PPK is almost always bilateral and not so rarely leads to visual impairment. This has an important consequence for the quality of life of the patient. MGOS encompasses a distinct MG-associated disease manifestation: our report aims to increase awareness for this underdiagnosed, potentially serious paraprotein-mediated consequence. 

For example, reduced visual acuity made our 57-year-old patient unable to work. His hematological disease would not cause any limitation in life without a visual impairment.

Minor comments:

1) Figure1 is not needed.

Our answer: It is required by the journal. Please see our modification.

2) It is better to show patient characteristics in a table.

Our answer: We added table 1

3) Most of the results about hematological evaluation are missing.

Our answer: We added table 2

4) lymphoplasmocytoid immunocytoma (line 223) ⇒ lymphoplasmacytoid immunocytoma

Our answer: Thank for the correction. We changed it to lymphoplasmacytic lymphoma. 

5) BJP type can be diagnosed by urine sample examination.

Our answer: Yes, it´s true. Examination of urine was part of the hematological workup. Since the deposits in the cornea may have been caused by the paraproteins in the serum, we mainly considered the serological findings during the evaluations.

Reviewer #4: The authors report corneal findings in patients diagnosed with monoclonal gammopathy with different disease severities such as MGUS, smoldering MM, and MM investigating the prevalence of paraproteinemic keratopathy (PPK) and involvement of specific proteins. They included 51 patients from hematology clinics with monoclonal gammopathy and evaluated them for corneal findings per their study protocol. They found that 19/51 patients had PPK, indicating 37% prevalence, which is higher than previously reported. They also found that FLC kappa was higher but not statistically significant in patients with PPK vs. without PPK, and FLC lambda was significantly lower in patients with PPK vs. without PPK. The study and results are relevant. They need to clarify some points in their methods and revise results and discussion accordingly.

Our comment: The primary aim of the study was to find out the prevalence of paraproteinemic keratopathy in patients suffering from monoclonal gammopathy. The 45 patients from group B first received the hematological diagnosis of the monoclonal gammopathy and later on received the ophthalmological evaluation. We assessed the prevalence of the paraproteinemic keratopathy based on the results from this group (13/45; prevalence of 29%).

The additional 6 patients (group A) first received the ophthalmological diagnosis of the paraproteinemic keratopathy. At this time the hematological diagnosis wasn´t even known. The hematological workup was initiated on ophthalmologist´s request and confirmed the diagnosis of monoclonal gammopathy in all cases. 

We decided to include these 6 patients (group A) to our analysis, but not to the prevalence assessment. We added this information in the methods section in lines 121-134 (version with mark-ups) 

Overall comments:

1. The authors state some correlations or associations they found or did not find throughout the text; however, no analysis was performed to check for correlations or associations. They need to revise wording throughout the manuscript, including the title.

Our answer: Thank you for your comment. We also performed an association test (Spearman's correlation) and found no correlation, so we revise the wording throughout the manuscript, including the title. This was our mistake, and we apologize for this.

2. Methods need some clarification about the recruitment and application of study procedures. Patients were probably recruited from the hematology department based on the inclusion/exclusion criteria, but it is unclear. It should be clarified where the study procedures were conducted and in which order. Hematological and ophthalmological evaluation subheadings require further detail in this regard. After the consent was obtained, what procedures were performed in which order, whether the blood test was performed on the same day or within a specific interval with the ophthalmologic evaluation.

Our answer: The majority of the study participants (n = 45, group B) were recruited at the hematology department with a confirmed diagnosis of monoclonal gammopathy. The ophthalmological examination followed the complete hematological workup and was performed within 3 months after the diagnosis of MG. 

However, there were additional six patients (group A), who first received the ophthalmological diagnosis of the paraproteinemic keratopathy. At this time the hematological diagnosis wasn´t even known. Actually, in group A the hematological workup was initiated on ophthalmologist´s request and was carried out within 3 months. 

We decided to include these six patients (group A) to our analysis, but not to the prevalence assessment. That is why we divided the cohort into two groups. 

We added this information in lines 121-134 (version with mark-ups) in the Methods section and hope to have clarified this point. 

3. Recommending having three study groups based on the classification of the hematologic disease. Then, they can have two subgroups based on the presence or absence of PPK to compare hematologic measures. It does not make sense to have groups A and B based on PPK was known or unknown as it does not help address any question.

Our answer: Thank you for your recommendation. We added table 1 for clarification. Although we considered having three study groups based on the classification of the hematological disease, it will not help us in investigating the correlation, as the number of study participants in these subgroups is very different (e.g. SMM n = 5 while MM n = 27). 

We divided our participants into 2 groups (A-ophthalmological diagnosis first; B- hematological diagnosis first) for the purpose of the prevalence assessment. When determining the prevalence, we had to exclude those patients for whom PPK was known before the MG, otherwise the prevalence would be incorrect.

Furthermore, we wanted to show the significance of the proper ophthalmological assessment, which in case of these six patients led to the hematological diagnosis of MG. 

Specific comments:

1. The abstract purpose may need to be rephrased to determine the prevalence of PPK among patients with MG.

Our answer: Thank you. We edited it. We hope to have clarified your concerns with the corrections in the methods section.

2. Abstract methods or results would consider adding n for each group as MGUS, SMM, and MM.

Our answer: Thank you. We added it.

3. According to this study, the prevalence of PPK in patients with MG was 37% (19/51), not 29%.

Our answer: As we explained in the previous comment, the prevalence was assessed only in 45 patients (group B) with newly confirmed MG and without any previously known corneal pathology (13/45). 

Six patients (group A) with the known diagnosis of PPK were not included in the prevalence assessment.

4. Lines 120-130: Further details needed regarding hematologic measures, when and where the laboratory test was performed? At a research lab or a certified clinical lab, where exactly? What were the unit measures?

Our answer: We added the information you suggested. The hematologic measures were performed according to the clinical routine of the Department of Hematology in the central laboratory of the Medical University Center of the Johannes Gutenberg University Mainz (lines 167-168). The units of measurement for the immunoglobulins and light chains are mentioned in the results section lines 263-271 (version with mark-ups).

5. Lines 134-140: Further details are needed regarding ophthalmologic evaluations. In what order were the procedures performed? How was the VA reported? Was Snellen converted to logMAR for analysis or not? Slit-lamp biomicroscopy and fundoscopy were explicitly performed to assess what details? What were parameters from tomography and OCT assessed?

Our answer: We added the information according to your suggestions in lines 179-186.

6. CONSORT flowchart in its current version does not seem helpful to give sufficient detail for this study. As this was a clinical trial, the CONSORT flowchart may not need to be used here. A revised flowchart can be helpful, though, to clarify the points mentioned above.

Our answer: We modified the flowchart. We cannot remove it completely due to the journal´s requirements.

7. Lines 156-158: A table with demographic and clinical characteristics of study participants would be helpful.

Our answer: We added a table as suggested (table 1).

8. Lines 159 and 162: It is confusing what these study groups refer to. How are these 6 patients different from 45 patients included in group B? If there were study groups, they should be defined in the Methods section first. If any subgroup analysis was planned, again, it should be described in the Methods section first. Authors may consider having 3 study groups based on the hematologic diagnosis as MGUS, SMM, and MM. See above for further details.

Our answer: Please see our answer to the overall comment no. 3. We added an explanation in the manuscript in the methods section. 

9. Lines 163-165: PPK in 19/51, 37% prevalence, among all participants is as important. A second table displaying the results between 163-170 (and maybe other results) according to the hematologic condition would be helpful.

Our answer: Please see our answers to the overall comment no. 3 and to the specific comment no. 3. We added table 2 in the results section.

10. Figure 3 does not show a correlation. Legend needs revision and a better description of what was displayed.

Our answer: Thank you for your suggestion. We revised the legend and added table 2 for better understanding.

11. Lines 193-201: Another table including demographic and clinical characteristics of patients according to PPK status would be helpful.

Our answer: We added tables 1 and 2.

12. Line 216: Were all these patients followed up or determined whether they needed PK based on the initial assessment, or only these two?

Our answer: The need for PK was assessed on the initial visit. The results of the follow-up will be a matter of further publication. PK was needed just in these 2 cases due to the decrease of vision.

13. Lines 219-231: How long did the study require to follow up of patients? What parameters were checked during the follow-up? Did they find any of the participants with no PPK at baseline developed PPK later?

Our answer: We aim to follow up the patients for 12 months (follow-ups after 3 months/6 months/12 months). The follow up examination include the complete ophthalmological workup as in the first evaluation as well as a complete laboratory examination including serum protein electrophoresis and immunofixation. So far no new PPK cases were observed in the follow up. The results of the longer follow-up will be evaluated.

14. Line 263-265: Consider including the reference paper that proposed MGOS terminology.

Our answer: We mentioned it in lines 88-89 included it as ref. no. 16

15. Line 339-340: “We recommend using the term MGOS for such patients” should be followed by “as proposed earlier” and relevant reference should be included.

Our answer: We go with it in line 453.

---

## [Decision Letter · Decision Letter 1]

6 May 2022

PONE-D-21-09281R1Interrelation between corneal and hematological findings in monoclonal gammopathyPLOS ONE

Dear Dr. Wasielica-Poslednik,

Thank you for submitting your manuscript to PLOS ONE. After careful consideration, we feel that it has merit but does not fully meet PLOS ONE’s publication criteria as it currently stands. Therefore, we invite you to submit a revised version of the manuscript that addresses the points raised during the review process.

We would like to apologies for the delays that have incurred on your submission. The revised manuscript has been re-reviewed by the reviewers and their comments may be seen below.

While it may be seen that one of the reviewers believe that additional language copy editing is required on the manuscript. During the internal evaluation of the manuscript we believe that the language is at a sufficient quality which allows the scientific understanding of information provided. Therefore we do not believe that  copy editing is required.

Furthermore as the reviewers agree that the current submission does not fall within the scope of a clinical trial, we would recommend that the CONSORT flow diagram is removed as to avoid confusion. It is also not necessary to provide the study protocols as supporting file and we recommend that this is also removed to avoid publication of these documents.

Finally, could you please carefully revise the manuscript to address all comments raised?

We look forward to receiving your revised manuscript.

Kind regards,

Lucinda Shen, MSc

Staff Editor

PLOS ONE

Reviewers' comments:

Reviewer's Responses to Questions

**Comments to the Author**

1. If the authors have adequately addressed your comments raised in a previous round of review and you feel that this manuscript is now acceptable for publication, you may indicate that here to bypass the “Comments to the Author” section, enter your conflict of interest statement in the “Confidential to Editor” section, and submit your "Accept" recommendation.

Reviewer #1: All comments have been addressed

Reviewer #2: All comments have been addressed

Reviewer #4: (No Response)

2. Is the manuscript technically sound, and do the data support the conclusions?

Reviewer #1: (No Response)

Reviewer #2: Yes

Reviewer #4: Partly

3. Has the statistical analysis been performed appropriately and rigorously? 

Reviewer #1: (No Response)

Reviewer #2: Yes

Reviewer #4: No

4. Have the authors made all data underlying the findings in their manuscript fully available?

Reviewer #1: (No Response)

Reviewer #2: Yes

Reviewer #4: (No Response)

5. Is the manuscript presented in an intelligible fashion and written in standard English?

Reviewer #1: (No Response)

Reviewer #2: Yes

Reviewer #4: (No Response)

6. Review Comments to the Author

Reviewer #1: (No Response)

Reviewer #2: Authors have made sufficient amendments to the manuscript. Table 1 should provide summarizing data of all patients and not only subgroups.

Reviewer #4: The authors revised their manuscript and addressed some of the concerns posed by the reviewers. However, the manuscript still needs extensive revisions. Even though the language and grammar are mostly ok, the manuscript is not well organized. They may benefit from a professional editing service to better construct the manuscript.

Overall comments:

1. This study is not a clinical trial. If it were, there would be an intervention investigated. What was the intervention investigated here? This is a prospective observational cross-sectional study. Therefore, its category should be changed, and then CONSORT will be irrelevant and not mandated by the journal.

2. Authors should be very clear and consistent about what their hypothesis was, what primary and secondary aims they had to conduct this study, and what main outcomes were selected. At the end of the introduction, the authors state, “The aim of the present study was to investigate whether the type and severity of the MG determines the presence, morphology, and severity of the PPK.” They should stick with this throughout the text, and their methods and results should reflect this. Their rebuttal letter says that the primary aim was to find out the prevalence. Which one? Their main outcome(s) should be determined based on that.

3. Study groups are usually formed to compare two or more groups regarding main outcomes to address the primary and secondary aims. The way the authors presented groups A and B is not scientifically acceptable. Participants included in each group represent two different cohorts. The authors chose to include group A into the analysis when they see the fit and exclude, which is a critical flaw. Participants in group A and group B cannot be expected to make a cohort as they were enrolled based on different characteristics. The statistical analysis would be biased if some patients were screened due to having a hematologic diagnosis and found to have a corneal disease, and some were seen in ophthalmology clinics maybe due to visual symptoms or diagnosed incidentally. These two groups cannot be merged for some part of the analysis conducted here. They should exclude those additional 6 patients and rearrange the whole manuscript accordingly. Their justification for including those 6 patients and then excluding them from the analysis is not scientifically convincing.

4. This study is conducted at one-time point. Cross-sectional. It was not a cohort study with long-term follow-up. They did not mention any follow-up in their methods. They did not mention any treatment (surgical or systemic) in their methods as a study intervention either. They cannot select two patients and talk about their surgical treatments and results. They also cannot report the results of the systemic treatment in one of them. The authors squeezed two case reports at the end of the results, which is not right. The study design should have been different if they were investigating the efficacy and safety of a surgical intervention (PK) and/or systemic treatment. They stated in their rebuttal letter that they are preparing another publication to report follow-up data. This would belong to that manuscript. They should remove it from this manuscript.

5. In their rebuttal letter, some of the points were not addressed; however, they responded as if they had made changes. For instance, they said they added n for MGUS, SMM, and MM in the abstract, but it was not added. Another table including demographic and clinical characteristics according to PPK status was suggested, and they said it was added, but there is no such table.

Specific comments:

Line 113-114. “primary diagnosis of PPK without any known hematologic disorder” - PPK diagnosis cannot be done without known hematologic disorder and/or cornea biopsy.

Lines 174-178. More details are needed re: histological evaluation. What tissue samples did they mean here? How were they collected?

Lines 180 to 191: Was a normality test performed? Which one? What were the main outcomes?

Line 197: Were both eyes included in the analysis? Please clarify. Data obtained from the right and left eyes are correlated and not recommended to be included in the analysis. Either a factor analysis would be necessary to account for this effect or another appropriate analysis method. Most commonly, only one eye would be included in the analysis. Authors are expected to clarify this in the methods section.

Table 1. MGUS, SMM, and MM should make the columns. Units of measurements should be added in relevant areas. Should include whether those results within the age columns are mean and SD or something else. As described in the statistics section, they should report the frequency of the categorical values.

Line 247 and 257. Recommend presenting in a table. This was suggested before, and the authors responded as if they did.

Lines 271 to 290. This paragraph does not belong to this paper as suggested above. This is not a longitudinal study, no follow up was planned for this study. Should exclude from this paper and include in the follow-up paper they mentioned. If they want to mention the necessity of PK in two patients, they can include a variable/parameter as PK recommended yes/no and discuss it but cannot include PK results during the follow up and discuss.

Lines 314-330: Can discuss that two patients had a significant corneal disease and were recommended PK but cannot discuss long-term treatments, systemic treatments given to these patients as cannot be reported as part of this study.

336-340: “This is probably due to the fact that the “non-crystalline” PPK types were unknown and subsequently overlooked or misdiagnosed. The most common form of PPK in the present study is a discreet stromal haze, which can easily be overlooked. The prevalence of crystalline keratopathy in our study was 2% (1 of 45 patients) and, hence, comparable with the prevalence found by Bourne at al.” and the following paragraph are essential points of this study.

Lines 373-381: They mention in statistical methods that they did the Spearman correlation test. No correlation results were reported in the result section. Then, they discuss here some results. They use different terms almost interchangeably, such as no association, statistically significant risk factor, and correlation, and report some p values. Were these reported in results? What statistical methods revealed these results exactly?

Lines 441-443: “We consider systemic therapy necessary in case of MGOS, especially before the penetrating keratoplasty. Nevertheless, a lifelong follow-up after corneal transplantation is necessary to rule out a corneal recurrence.” Not the result of this study, cannot conclude that based on this study findings. Should consider removing.

7. PLOS authors have the option to publish the peer review history of their article (what does this mean?). If published, this will include your full peer review and any attached files.

Reviewer #1: No

Reviewer #2: **Yes: **Linus Voelker

Reviewer #4: No

---

## [Author Response · Author response to Decision Letter 1]

24 Jun 2022

PONE-D-21-09281

Prevalence of corneal findings and their interrelation with hematological findings in monoclonal gammopathy

Dear Editor,

we thank you and the reviewers for the valuable comments and suggestions and hope to have improved our manuscript now. Below you will find our responses to the editor´s and reviewers´ comments.

With kindest regards

Mohammad Al Hariri

Joanna Wasielica-Poslednik

Reviewers' comments:

Reviewer's Responses to Questions

Comments to the Author

1. If the authors have adequately addressed your comments raised in a previous round of review and you feel that this manuscript is now acceptable for publication, you may indicate that here to bypass the “Comments to the Author” section, enter your conflict of interest statement in the “Confidential to Editor” section, and submit your "Accept" recommendation.

Reviewer #1: All comments have been addressed

Reviewer #2: All comments have been addressed

Reviewer #4: (No Response)

Our answer: thank you for your interest

2. Is the manuscript technically sound, and do the data support the conclusions?

Reviewer #1: (No Response)

Reviewer #2: Yes

Reviewer #4: Partly

Our answer: thank you for your evaluation

3. Has the statistical analysis been performed appropriately and rigorously?

Reviewer #1: (No Response)

Reviewer #2: Yes

Reviewer #4: No

Our answer: thank you for your evaluation

4. Have the authors made all data underlying the findings in their manuscript fully available?

Reviewer #1: (No Response)

Reviewer #2: Yes

Reviewer #4: (No Response)

5. Is the manuscript presented in an intelligible fashion and written in standard English?

Reviewer #1: (No Response)

Reviewer #2: Yes

Reviewer #4: (No Response)

6. Review Comments to the Author

Reviewer #1: (No Response)

Reviewer #2: Authors have made sufficient amendments to the manuscript. Table 1 should provide summarizing data of all patients and not only subgroups.

Our answer: thank you for your recommendation. We added summarizing data of all patients.

Reviewer #4: The authors revised their manuscript and addressed some of the concerns posed by the reviewers. However, the manuscript still needs extensive revisions. Even though the language and grammar are mostly ok, the manuscript is not well organized. They may benefit from a professional editing service to better construct the manuscript.

Our answer: thank you for your comment. We strive to improve our manuscript and thank you for your suggestions for improvement.

Overall comments:

1. This study is not a clinical trial. If it were, there would be an intervention investigated. What was the intervention investigated here? This is a prospective observational cross-sectional study. Therefore, its category should be changed, and then CONSORT will be irrelevant and not mandated by the journal.

Our answer: thank you for your comment. In fact, our study is an observational cross-sectional study without intervention. We changed this category in the methods section and removed the CONSORT flow diagram as recommended.

2. Authors should be very clear and consistent about what their hypothesis was, what primary and secondary aims they had to conduct this study, and what main outcomes were selected. At the end of the introduction, the authors state, “The aim of the present study was to investigate whether the type and severity of the MG determines the presence, morphology, and severity of the PPK.” They should stick with this throughout the text, and their methods and results should reflect this. Their rebuttal letter says that the primary aim was to find out the prevalence. Which one? Their main outcome(s) should be determined based on that.

Our answer: thank you for this comment. We agree that the aims and hypotheses of the study were not clearly defined in the previous versions of the manuscript. Our primary aim was to determine the prevalence of PPK in the MG population. Our secondary aim was to investigate whether the type and severity of the MG determines the presence, morphology, and severity of the PPK. 

Therefore, we changed the title; reformulated the purpose in the abstract; defined the aims and hypotheses in the last paragraph of the introduction; added titles of the subsections in the results section; and discuss our results according to the aims and hypotheses in the first two paragraphs of the discussion. 

3. Study groups are usually formed to compare two or more groups regarding main outcomes to address the primary and secondary aims. The way the authors presented groups A and B is not scientifically acceptable. Participants included in each group represent two different cohorts. The authors chose to include group A into the analysis when they see the fit and exclude, which is a critical flaw. Participants in group A and group B cannot be expected to make a cohort as they were enrolled based on different characteristics. The statistical analysis would be biased if some patients were screened due to having a hematologic diagnosis and found to have a corneal disease, and some were seen in ophthalmology clinics maybe due to visual symptoms or diagnosed incidentally. These two groups cannot be merged for some part of the analysis conducted here. They should exclude those additional 6 patients and rearrange the whole manuscript accordingly. Their justification for including those 6 patients and then excluding them from the analysis is not scientifically convincing.

Our answer: thank you for this comment. We created groups A and B for better understanding of the origin of our patients. Group A consisted of patients with the primary ophthalmological suspicion of PPK (and a confirmed diagnosis of MG) and group B consisted of patients with the primary hematological diagnosis of MG. 

 We agree that the groups were not acceptable. Therefore, we deleted “groups A and B” throughout the manuscript. We modified the Methods section (lines 124-133) and highlighted that the prevalence was evaluated only in patients with the primary hematological diagnosis (Methods lines 127-128). In the Results subsection “Prevalence of PPK”, we added table 2 considering only 45 patients recruited in the Department of Hematology.

However, the patients recruited at the Department of Ophthalmology (primary PPK suspicion) fulfill the in- and exclusion criteria of the study as they were diagnosed with MG and provide many valuable information regarding the secondary aim of the study (interrelation between corneal and hematological findings). Therefore, we don´t want to exclude them from this part of the study and to lose valuable information on this rare disease. We added this information in the Methods section (lines 129-133) and in the Results section (lines 217-220).

4. This study is conducted at one-time point. Cross-sectional. It was not a cohort study with long-term follow-up. They did not mention any follow-up in their methods. They did not mention any treatment (surgical or systemic) in their methods as a study intervention either. They cannot select two patients and talk about their surgical treatments and results. They also cannot report the results of the systemic treatment in one of them. The authors squeezed two case reports at the end of the results, which is not right. The study design should have been different if they were investigating the efficacy and safety of a surgical intervention (PK) and/or systemic treatment. They stated in their rebuttal letter that they are preparing another publication to report follow-up data. This would belong to that manuscript. They should remove it from this manuscript.

Our answer: thank you for recommendation. We removed this part from the manuscript. 

5. In their rebuttal letter, some of the points were not addressed; however, they responded as if they had made changes. For instance, they said they added n for MGUS, SMM, and MM in the abstract, but it was not added. Another table including demographic and clinical characteristics according to PPK status was suggested, and they said it was added, but there is no such table.

Our answer: we sincerely apologize for these unintentional mistakes. This time, we surely added n for MGUS, SMM and MM in the abstract. We added table 1 with demographic and clinical characteristics according to PPK of all patients in the Results section. 

Specific comments:

Line 113-114. “primary diagnosis of PPK without any known hematologic disorder” - PPK diagnosis cannot be done without known hematologic disorder and/or cornea biopsy.

Our answer: thank you for this comment. We change it to primary suspicion.

Lines 174-178. More details are needed re: histological evaluation. What tissue samples did they mean here? How were they collected?

Our answer: “Histological evaluation” was part of description of the surgical cases. The whole part was now removed on your request.

Lines 180 to 191: Was a normality test performed? Which one? What were the main outcomes?

Our answer: we did not perform a normality test, but we checked the distribution in the histogram. The data in the histogram were not normally distributed.

Line 197: Were both eyes included in the analysis? Please clarify. Data obtained from the right and left eyes are correlated and not recommended to be included in the analysis. Either a factor analysis would be necessary to account for this effect or another appropriate analysis method. Most commonly, only one eye would be included in the analysis. Authors are expected to clarify this in the methods section.

Our answer: Both eyes were included in the analyses. Ocular findings were the same in both eyes in all patients. In the analysis of ocular involvement in MG, either both eyes were involved or not. Thus, we decided to include the mean value of right and left eyes into the analysis, which is a common procedure if always both eyes are affected in an individual subject. We have included this in the methods section (lines 184-187). In the analysis of visual acuity, corneal thickness, and densitometry, involved vs. uninvolved eyes were examined. 

Table 1. MGUS, SMM, and MM should make the columns. Units of measurements should be added in relevant areas. Should include whether those results within the age columns are mean and SD or something else. As described in the statistics section, they should report the frequency of the categorical values.

Our answer: As you previously wished we rearranged table 1 to be according to PPK status, made MGUS, SMM and MM to columns and included the information of age frequency of the categorical values in the table.

Line 247 and 257. Recommend presenting in a table. This was suggested before, and the authors responded as if they did.

Our answer: we apologize for this misunderstanding. The requested table is now provided.

Lines 271 to 290. This paragraph does not belong to this paper as suggested above. This is not a longitudinal study, no follow up was planned for this study. Should exclude from this paper and include in the follow-up paper they mentioned. If they want to mention the necessity of PK in two patients, they can include a variable/parameter as PK recommended yes/no and discuss it but cannot include PK results during the follow up and discuss.

Our answer: Removed

Lines 314-330: Can discuss that two patients had a significant corneal disease and were recommended PK but cannot discuss long-term treatments, systemic treatments given to these patients as cannot be reported as part of this study.

Our answer: Removed

336-340: “This is probably due to the fact that the “non-crystalline” PPK types were unknown and subsequently overlooked or misdiagnosed. The most common form of PPK in the present study is a discreet stromal haze, which can easily be overlooked. The prevalence of crystalline keratopathy in our study was 2% (1 of 45 patients) and, hence, comparable with the prevalence found by Bourne at al.” and the following paragraph are essential points of this study.

Our answer: thank you! We do think that PPK is mostly a very discreet finding – “crystals” may be seen in the confocal microscopy, but not on a slit-lamp. We would like to work on it the next paper..

Lines 373-381: They mention in statistical methods that they did the Spearman correlation test. No correlation results were reported in the result section. Then, they discuss here some results. They use different terms almost interchangeably, such as no association, statistically significant risk factor, and correlation, and report some p values. Were these reported in results? What statistical methods revealed these results exactly?

Our answer: Thank you for this correction. Initially, we planned to perform Spearman correlation, but as the PPK status is a dichotomous variable, Mann-Whitney-U testing showed the same information indicating no association between the level of paraprotein and disease status. 

We removed Spearman correlation from the “statistical analysis” and modified some parts of the discussion (please see lines 470-502). 

Lines 441-443: “We consider systemic therapy necessary in case of MGOS, especially before the penetrating keratoplasty. Nevertheless, a lifelong follow-up after corneal transplantation is necessary to rule out a corneal recurrence.” Not the result of this study, cannot conclude that based on this study findings. Should consider removing.

Our answer: We removed this paragraph.

7. PLOS authors have the option to publish the peer review history of their article (what does this mean?). If published, this will include your full peer review and any attached files.

Do you want your identity to be public for this peer review? For information about this choice, including consent withdrawal, please see our Privacy Policy.

Reviewer #1: No

Reviewer #2: Yes: Linus Voelker

Reviewer #4: No

While revising your submission, please upload your figure files to the Preflight Analysis and Conversion Engine (PACE) digital diagnostic tool, https://pacev2.apexcovantage.com/. . PACE helps ensure that figures meet PLOS requirements. To use PACE, you must first register as a user. Registration is free. Then, login and navigate to the UPLOAD tab, where you will find detailed instructions on how to use the tool. If you encounter any issues or have any questions when using PACE, please email PLOS at figures@plos.org. Please note that Supporting Information files do not need this step.

---

## [Decision Letter · Decision Letter 2]

1 Sep 2022

PONE-D-21-09281R2Prevalence of corneal findings and their interrelation with hematological findings in monoclonal gammopathyPLOS ONE

Dear Dr. Wasielica-Poslednik,

Thank you for submitting your manuscript to PLOS ONE. After careful consideration, we feel that it has merit but does not fully meet PLOS ONE’s publication criteria as it currently stands. Therefore, we invite you to submit a revised version of the manuscript that addresses the points raised during the review process.

We look forward to receiving your revised manuscript.

Kind regards,

Homayon Ghiasi, PhD

Academic Editor

PLOS ONE

Journal Requirements:

Reviewers' comments:

Reviewer's Responses to Questions

**Comments to the Author**

1. If the authors have adequately addressed your comments raised in a previous round of review and you feel that this manuscript is now acceptable for publication, you may indicate that here to bypass the “Comments to the Author” section, enter your conflict of interest statement in the “Confidential to Editor” section, and submit your "Accept" recommendation.

Reviewer #1: All comments have been addressed

Reviewer #5: (No Response)

2. Is the manuscript technically sound, and do the data support the conclusions?

Reviewer #1: (No Response)

Reviewer #5: Partly

3. Has the statistical analysis been performed appropriately and rigorously? 

Reviewer #1: (No Response)

Reviewer #5: Yes

4. Have the authors made all data underlying the findings in their manuscript fully available?

Reviewer #1: (No Response)

Reviewer #5: Yes

5. Is the manuscript presented in an intelligible fashion and written in standard English?

Reviewer #1: (No Response)

Reviewer #5: Yes

6. Review Comments to the Author

Reviewer #1: (No Response)

Reviewer #5: The aim of the study was to determine prevalence of paraproteinemic keratopathy among patients with monoclonal gammopathy, and to evaluate possible interrelation between corneal and hematological parameters.

The findings are of interest, and they are put in the context of previous publications. The authors have addressed the comments raised in the previous round of review.

Still, there are couple of points that should be corrected or described more clearly:

Abstract:

r.40-41 The conclusion that „Median level of free light-chain 40 (FLC) kappa in serum of patients with kappa-restricted plasma cell dyscrasia was higher in 41 patients with PPK compared to patients without PPK (p=0.18)“ is not correct because the p value does not reach the level of significance

Methods and Results:

The „PPK prevalence and morphology“ is described first in the“ Results section, then the „Hematological findings“, and finally the „Ophthalmological findings“. However, The PPK prevalence and morphology belongs to the Ophthalmological findings, too. The results should be presented more systematically.

Was the IVCM examination performed in patients with diagnosis of PPK, only? In the view the densitometry results were not different between the patients with and without PPK, and the fact, the previous studies have found a difference in densitometry between the MG patients and age-adjusted controls, it could be possible that some pathological findings could be proven even in the „non-PPK“ patients using IVCM! This should also be mentioned in the discussion along the fact the control group is missing.

In the Methods section, there is indicated that tonometry, fundoscopy, and macular OCT was also performed to rule out the other ocular involvement , but no outputs are mentioned in the Results section.

Discussion:

r.331-332 The prevalence of crystalline keratopathy is presented. However, this is not described in the „Results section“ earlier. Also, check the form „crystalline“ is used all over the text, not „cristalline“ (e.g. r.336 „non-cristalline“)

7. PLOS authors have the option to publish the peer review history of their article (what does this mean?). If published, this will include your full peer review and any attached files.

Reviewer #1: No

Reviewer #5: No

---

## [Author Response · Author response to Decision Letter 2]

27 Sep 2022

Mainz, 27.09.22

PONE-D-21-09281_R3

Prevalence of corneal findings and their interrelation with hematological findings in monoclonal gammopathy

Dear Editor,

we thank you and the reviewers for the valuable comments and suggestions and hope to have improved our manuscript now. Below you will find our responses to the reviewers´ comments.

With kindest regards

Mohammad Al Hariri

Joanna Wasielica-Poslednik

Reviewer #5: The aim of the study was to determine prevalence of paraproteinemic keratopathy among patients with monoclonal gammopathy, and to evaluate possible interrelation between corneal and hematological parameters.

The findings are of interest, and they are put in the context of previous publications. The authors have addressed the comments raised in the previous round of review.

Still, there are couple of points that should be corrected or described more clearly:

Abstract:

r.40-41 The conclusion that „Median level of free light-chain 40 (FLC) kappa in serum of patients with kappa-restricted plasma cell dyscrasia was higher in 41 patients with PPK compared to patients without PPK (p=0.18)“ is not correct because the p value does not reach the level of significance

Thank you for this comment. We changed this part of the abstract (lines 38-42).

Methods and Results:

The „PPK prevalence and morphology“ is described first in the“ Results section, then the „Hematological findings“, and finally the „Ophthalmological findings“. However, The PPK prevalence and morphology belongs to the Ophthalmological findings, too. The results should be presented more systematically.

Thank you for this suggestion. We changed the name „ocular findings“ to „clinical findings“ and swapped the order of the last two parts of the results. We hope to have improved the understanding.

Was the IVCM examination performed in patients with diagnosis of PPK, only? In the view the densitometry results were not different between the patients with and without PPK, and the fact, the previous studies have found a difference in densitometry between the MG patients and age-adjusted controls, it could be possible that some pathological findings could be proven even in the „non-PPK“ patients using IVCM! This should also be mentioned in the discussion along the fact the control group is missing.

IVCM examination was performed in all patients as it was a prospective study – we added this information in the first sentence of the „Ophthalmological evaluation“ section (line 166). 

We found hyperreflective needle-like crystalline stromal deposits using IVCM in all corneas with the diagnosis of PPK on the slit lamp examination (Results section/IVCM lines 250-251).

In addition, 18 other patients had less pronounced hyperreflective point- and/or needle-like deposits in the deep stroma without any discernible opacities in the morphological examination of the cornea using the slit lamp. We added this information in the Results section/IVCM (lines 252-254).

We added the following sentence in the discussion (lines 378-381): “Furthermore, using IVCM we observed hyperreflective point- and/or needle-like deposits in the deep stroma of patients without any PPK-like opacities visible on slit lamp. Future clinical trial with a healthy control group should answer the question if IVCM could detect the “pre-slit-lamp” PPK forms.”

We agree that the limitation of the study is a lack of a control group. We mention it in the discussion (line 466). We plan to include a control group into further studies on this topic.

In the Methods section, there is indicated that tonometry, fundoscopy, and macular OCT was also performed to rule out the other ocular involvement , but no outputs are mentioned in the Results section.

Thank you for this advice. We added this information in the „Clinical Findings“ section.

One patient with Waldenstrom's disease had bilateral macular involvement in form of a subfoveal defect of the retinal pigment epithelium with cystoid macular edema and one patient with SMM type IgA λ had macular involvement in the form of a focal detachment of the neurosensory retina with secondary choroidal neovascularization (CNV). We interpret these findings as paraproteinemic maculopathy. We added these information in Clinical findings (lines 268-275).

Discussion:

r.331-332 The prevalence of crystalline keratopathy is presented. However, this is not described in the „Results section“ earlier. Also, check the form „crystalline“ is used all over the text, not „cristalline“ (e.g. r.336 „non-cristalline“)

Thank you for the remark. We added this information in the Methods section (lines 218-221).

We use term "crystalline" within the manuscript.

---

## [Editor Report · Decision Letter 3]

29 Sep 2022

Prevalence of corneal findings and their interrelation with hematological findings in monoclonal gammopathy

PONE-D-21-09281R3

Dear Dr. Wasielica-Poslednik<o:p></o:p>

We’re pleased to inform you that your manuscript has been judged scientifically suitable for publication and will be formally accepted for publication once it meets all outstanding technical requirements.

Kind regards,

Homayon Ghiasi, PhD

Academic Editor

PLOS ONE

Additional Editor Comments (optional):

Please revise the manuscript based on the recommendation of reviewer #3. Also we apologize for the long delay.
---

## [Editor Report · Acceptance letter]

3 Oct 2022

PONE-D-21-09281R3 

Prevalence of corneal findings and their interrelation with hematological findings in monoclonal gammopathy 

Dear Dr. Wasielica-Poslednik:

I'm pleased to inform you that your manuscript has been deemed suitable for publication in PLOS ONE. Congratulations! Your manuscript is now with our production department. 

Kind regards, 

on behalf of

Dr. Homayon Ghiasi 

Academic Editor

PLOS ONE